# Experimental Evaluation of Continuous In-Situ Biomethanation of CO$_2$ in Anaerobic Digesters Fed on Sewage Sludge and Food Waste and the Influence of Hydrogen Gas–Liquid Mass Transfer

**Davide Poggio [1],\*, Arman Sastraatmaja [1], Mark Walker [2], Stavros Michailos [2], William Nimmo [1] and Mohamed Pourkashanian [1],\***

1   Energy2050, Department of Mechanical Engineering, Faculty of Engineering, University of Sheffield, Sheffield S10 2TN, UK
2   School of Engineering, University of Hull, Hull HU6 7RX, UK
\*   Correspondence: d.poggio@sheffield.ac.uk (D.P.); m.pourkashanian@sheffield.ac.uk (M.P.)

**Abstract:** In-situ biomethanation combines conventional biogas production from the anaerobic digestion (AD) of organic matter with the addition of hydrogen to produce a higher quality biomethane gas. However, challenges surrounding its performance and control could hinder its uptake. To investigate this, an automated rig was designed and operated to study in-situ biomethanation with sewage sludge (SS) and food waste (FW) feedstocks. The effects that were experimentally investigated included the biogas recirculation rate, stirring intensity, and organic loading rate (OLR). All the results highlighted the rate-limiting effect of H$_2$ gas–liquid mass transfer (measured $k_L a$ in the range of 43–82 day$^{-1}$), which was implied by a lack of evidence of hydrogen-induced biological inhibition and a high average equilibrium hydrogen content in the biogas (a volume of 7–37%). At an OLR of 2 g VS L$^{-1}$day$^{-1}$, increasing biogas recirculation and mechanical stirring rates improved the methane evolution rate up to 0.17 and 0.23 L L$^{-1}$day$^{-1}$ and the H$_2$ conversion up to 80 and 66% for sewage sludge and food waste, respectively. A lower OLR of 1 g VS L$^{-1}$day$^{-1}$ allowed for increased hydrogen conversion but at a lower level of methane productivity. A process model, validated on experimental data, predicted that improving the $k_L a$ to at least 240 day$^{-1}$ would be required for in-situ biomethanation at OLRs common in AD systems in order to achieve a drop-in quality in terms of the biogas, with further downstream treatment required for certain applications.

**Keywords:** biomethanation; in-situ; biogas upgrading; hydrogen; gas–liquid mass transfer





## 1. Introduction

Interest in the methanation of hydrogen in a power-to-gas concept has increased, since methane currently has a broader range of drop-in applications where natural gas is the existing fuel [1] (e.g., in industrial and domestic heating, power generation, and vehicle fuel) and is therefore often preferred to hydrogen as an energy vector. The biological methanation of hydrogen and carbon dioxide can be applied in such a concept where electricity (e.g., produced by intermittent renewables) is used to produce hydrogen through electrolysis, which is then converted through the action of hydrogenotrophic methanogens [2] to produce biomethane in the following reaction:

$$CO_2 + 4H_2 \rightarrow CH_4 + 2H_2O \tag{1}$$

The biomethanation reaction can take place in either a dedicated reactor (ex-situ) or, as studied in this work, within anaerobic digesters (in-situ) [3,4] fed on both biomass and hydrogen, where the carbon dioxide requirement is satisfied by the biogas from the anaerobic digestion (AD) process itself. By this method, the methane content in the biogas can be increased from 60–70% to >90% [5].

Previous in-situ biomethanation studies mainly focussed on optimising the methane production or composition [6,7], studying the microbiology of the process [8], or identifying the limitations of the process [9]. Modelling has provided insight and supported evidence that biomethanation might be rate-limited by either biological or mass-transfer (i.e., $H_{2(g)} \rightarrow H_{2(aq)}$) processes [10] using a modified version of Anaerobic Digestion Model 1 (ADM1) [11], but of the two limitations, gas–liquid mass transfer is the most commonly cited issue in experimental biomethanation works [12,13].

The gas–liquid mass transfer characteristics of a continuously stirred tank reactor (CSTR) can be modified using different gas delivery methods (e.g., membranes [6] or diffusers [12,14]), using mechanical mixing [3,13], or biogas gas recirculation, with the latter having been demonstrated in ex- and in-situ systems at both laboratory and pilot scale [12,14–17] and shown to increase hydrogen consumption [15]. A broad variation in the gas recirculation rate has been investigated, with it ranging from 2.5 [15] to 240 $L_{gas}$ $L_{reactor}^{-1}$ $day^{-1}$ [14]. In general, it has been reported that higher recirculation rates increase the efficiency of hydrogen uptake in the biomethanation process [12,14,17]. Excessive recirculation could, however, hinder the process by causing turbulence and foaming and cause practical operational issues, in addition to the requirement of an unfeasible energy input for the gas pumping [17].

$CO_2$ conversion in in-situ biomethanation reduces the amount of dissolved inorganic carbon in the system, which is important due to its role in buffering the pH; its removal can cause elevated pH increase [6], leading to the inhibition of, in particular, acetoclastic methanogens [3] and the disruption of the underlying digestion of the feedstock. Therefore, the quantity of hydrogen introduced into the system is critical; too much can cause the inhibition of biological processes and the depletion of the carbonate buffer, whereas too little hydrogen injection can result in lower performance in terms of the produced biomethane quality. According to the biomethanation reaction (1), hydrogen should be introduced in a molar ratio of 4 with the desired carbon dioxide consumption [18,19]. However, the accurate estimation of the hydrogen requirement requires updated current knowledge of the background carbon dioxide production, which cannot be observed directly, and introducing the hydrogen alongside the AD process may also alter its performance [9,20].

In prior works studying continuous or semi-continuous in-situ biomethanation, process parameters such as gas composition were monitored on a daily basis, e.g., through the use of a gas bag in order to analyse initial and residual hydrogen on a daily basis [6], a process which was adapted in [9] with a gas recirculation loop that was also monitored on a daily basis. The limitations of such systems is that a relatively large variable volume of gas storage is required to achieve a high conversion of hydrogen before the product biogas can be discharged [21], with the configuration requiring substantial modifications in the operation of conventional AD reactors.

Building on previous studies, the objectives of this work were:

1. The development of an automated rig for in-situ biomethanation to emulate the full-scale implementation of biomethanation in a continuous AD plant, with control of the hydrogen injection rate based on feedback control and the online monitoring of process parameters.

2. The investigation of a selection of process design characteristics, i.e., biogas recirculation, biogas sparging, mechanical mixing, and the organic loading rate (OLR), on the performance of in-situ biomethanation.

3. The use of experimental findings to support model-based analysis of the implications for the realistic operation of continuous in-situ biomethanation.

## 2. Materials and Methods

### 2.1. Feedstock, Inoculums, and Trace Elements

Sewage sludge (SS) and food waste (FW) were used as feedstocks for the biomethanation experiments. Both feedstocks, after collection, transport, processing, and homogenisation were stored in a freezer at $-18\ °C$ in 2-litre batches to be thawed on demand. SS

consisted of mixed primary and secondary sludge and was collected from Stockport Waste Water Treatment Works. FW was collected as a single sample (~80 kg) of source segregated food waste from a university canteen. Contaminants (e.g., packaging) were removed, and the remaining organic sample was segregated according to the major categories of food waste and recombined in the suggested relative quantities of UK FW [22] to produce a final sample of ~40 kg and homogenised in a blender (Magimix 5200XL, Vincennes, France) and commercial mincer (Tritacarne No.12, Tre Spade, Torino, Italy). FW was diluted before use (to improve pump performance) by adding 70% (*w/w*) of DI water to reach a target total solids (TS) concentration of 14%.

To reduce the acclimatisation periods, the SS and FW experiments were inoculated using digestate from a full-scale SS digester (Stockport Wastewater Treatment Works, Stockport, UK) and a commercial FW digester (ReFood, Doncaster, UK), respectively. The inoculums were used on the same day as collection and were screened to ~1 mm before use.

To avoid any effect of trace nutrient limitation (which has been widely reported for the AD of FW [23,24]), trace elements were added to the feedstocks at a volumetric ratio of 1:1000 using the recommended composition and approach as per [20].

### 2.2. Analytical Methods

Standard analytical methods, where available, were used in this work. The total and volatile solids (TS, VS) were measured using a gravimetric approach, 2540 APHA [25], at temperatures of 105 °C and 550 °C, respectively. The alkalinity was measured using a titration method, according to APHA 2320 [26]. A pH endpoint of 5.7 was used for intermediate alkalinity (IA) and 4.3 was used for partial (PA) and total alkalinity (TA) [26]. The total ammonia nitrogen (TAN) was measured based on the standard method 4500 $NH_3$ B [25]. An elemental analysis was performed using a Flash 2000 Elemental Analyser (Thermo Fisher Scientific Inc., Waltham, MA, USA) as per the manufacturer's instructions, using vanadium pentoxide as a catalyst and BBOT (2,5- Bis (5-tert-butyl-benzoxazol-2-yl) thiophene) as a calibration standard. Samples for volatile fatty acids (VFAs) analysis were acidified with 5% formic acid (*v/v*), centrifuged (Heraeus™ Pico™ 21 Microcentrifuge, Thermo Fisher Scientific Inc., Waltham, MA, USA) for 30 min at 14,000 rpm, and filtered to 0.2 μm. VFAs were measured by gas chromatography (Thermo Scientific™ TRACE™ 1300, Thermo Fisher Scientific Inc., Waltham, MA, USA), using a flame ionisation (FID) detector and a DB-FFAP column with nitrogen as the carrier gas. Standards of 10 mM, 2.5 mM, and 1 mM of VFAs were used for calibration.

### 2.3. Biochemical Methane Potential Test

Biochemical methane potential (BMP) tests were carried out using an AMPTS II (Bioprocess Control AB, Lund, Sweden) instrument as per the manufacturer's instructions. Sewage and food waste-adapted inocula were used for the respective feedstocks. Triplicate measurements were made for blank (inoculum only), control (inoculum + cellulose) for both inoculums, and biomass (inoculum + feedstock) for both FW and SS, using the respective inoculum, totalling in 30 individual tests. The inoculum to substrate ratio (ISR) was selected at 3 on a VS basis. The test continued until daily methane production was less than 1% of the cumulative volume on three consecutive days [27]. Inoculum biological activity was validated by the measurement control substance (cellulose), which has a reported BMP in the range 354–370 $m^3$ $CH_4$ $g^{-1}$ VS [28].

### 2.4. In-Situ Biomethanation Experimental Setup

A laboratory-scale in-situ biomethanation experimental rig was designed and built, allowing for continuous semi-automated operation and online monitoring. A schematic of the main functional elements is shown in Figure 1, including the reactors, instruments, sensors, and SCADA (supervisory control and data acquisition) connections; a photo of the experimental setup is shown in Figure 2. This description is abridged, highlighting the

main design features. For a complete description including the design, commissioning, and initial testing of the equipment, see [29].

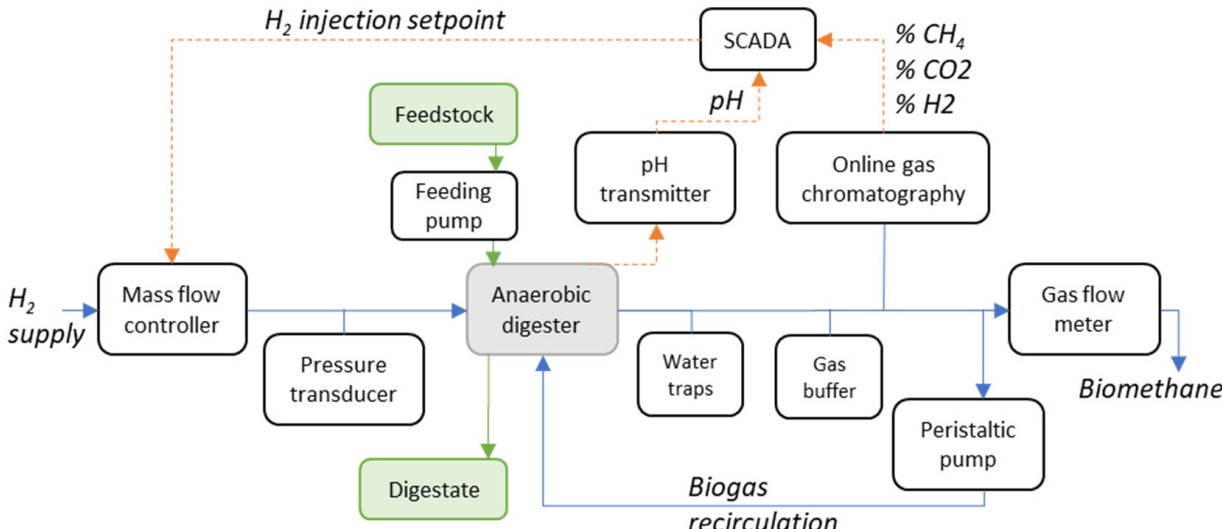

**Figure 1.** Schematic of the in-situ biomethanation experimental rig.

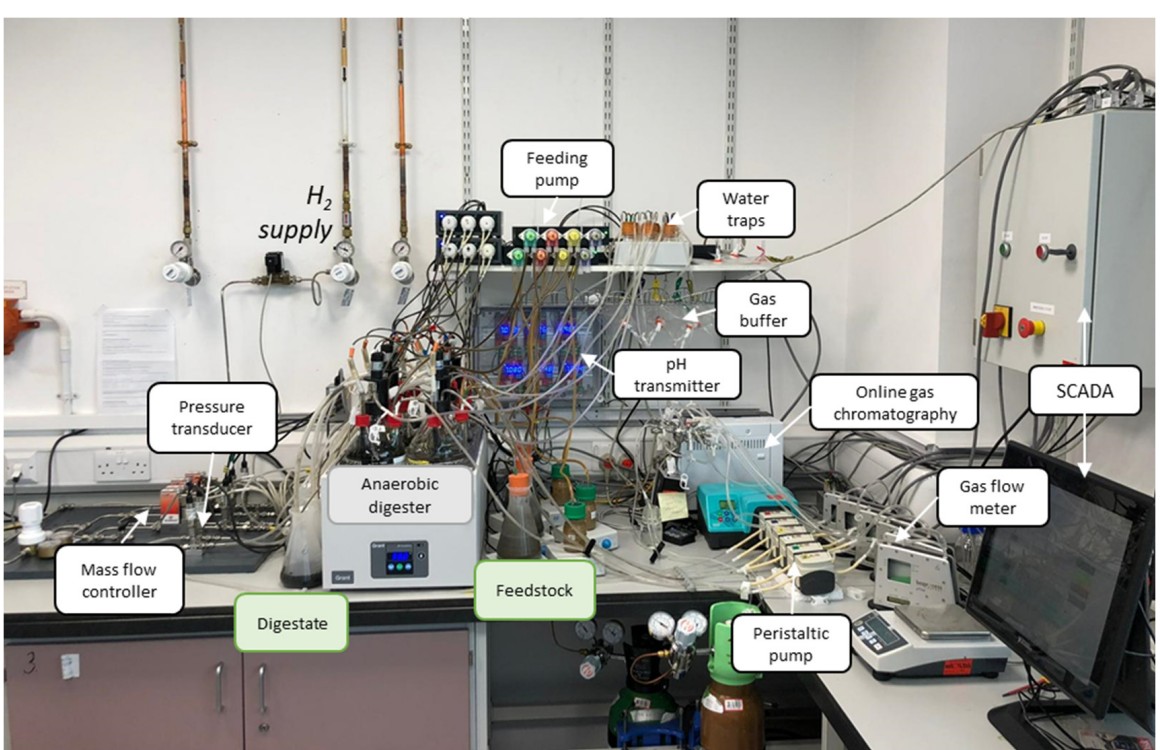

**Figure 2.** Photo of the in-situ biomethanation experimental rig with main components labelled.

The main features of the experimental setup were as follows.

Reactors: 6x CSTR with 2 L capacity (1.7 L working volume $V_R$), temperature control at 38 °C in a thermostatic water bath with biogas recirculation by peristaltic pump (323/D Watson Marlow, Falmouth, UK) at a recycling rate between 20 and 280 rpm (12 to 155 L $L_R^{-1}$ day$^{-1}$). The overall gas phase volume $V_H$ of the setup, including the reactors headspaces and the water traps volumes, was estimated at 0.6 L.

Hydrogen injection: controllable, automated, and independent (per reactor) hydrogen injection, from a pressurised cylinder, using mass flow controllers (MFC) (EL-Flow select,

Bronkhorst High-Tech B.V., Ruurlo, Netherlands), calibrated for hydrogen in the range of 0.16–0.8 mL min$^{-1}$ (measured error $3.3 \pm 0.4\%$) and introduced to each digester anaerobically via a Tygon tube (Saint Goibain, Courbevoie, France) to the gas port on top of the reactor and through a submerged stainless steel sparger with a 2 μm pore size.

Feedstock feeding: automated feedstock feeding and digestate removal, by programmable dosing pump (D-DH2Ocean P4 Pro, Kamoer, Shanghai, China) with a fixed setpoint, monitored daily by weighing. Feed for each reactor was prepared in a 0.5 litre feeding container and placed on a magnetic stirrer to maintain the homogenisation of feeding. SS was fed twelve times a day and FW four times a day.

A monitoring and control system (SCADA) was implemented to (1) ensure safe operation, (2) provide closed loop control of the hydrogen injection rate, and (3) monitor key experimental parameters, using a controller (CompactRIO™ cRIO-9045, National Instruments, Austin, TX, USA) consisting of analogue and digital modules, along with a custom LabVIEW™ application.

Safety features: the detection of pressure build-up in the reactors and hydrogen leak in the laboratory and a solenoid valve on the hydrogen injection system, for the automated safety shutoff of the rig.

### 2.5. In-Situ Biomethanation–Online Monitoring

The biogas composition was measured through an online gas chromatograph (490 Micro Gas Chromatograph, Agilent, Santa Clara, USA) and stream selector (VICI Valco$^{®}$ instrument, Houston, USA). The GC sampled each stream every 20 min with a single analysis taking 3 min. Prior to sampling, the line was flushed by the GC sampling pump for 30 s, from the reactor, to avoid cross contamination from other reactors; biogas recirculation ensures that the sample is characteristic of the headspace composition. A 200 mL gas bag was installed on each reactor to avoid the vacuum effect due to both liquid and gas sampling. The gas flushing and sampling volumes were accounted for in the calculation as gas produced. The GC had a dual cabinet equipped with a CP-Molsieve channel (Molsieve 5A PLOT 0.25 mm in, 20 m, Argon carrier, measures $H_2$, $O_2$, $N_2$, $CH_4$) and a CP-PoraPLOT U channel (PoraPLOT U, 0.25 mm, and 10 m, Helium carrier, measures $CO_2$, $H_2S$). The total biogas outflow was measured from each reactor using a gas flow meter (μFlow$^{TM}$ Bioprocess Control AB, Lund, Sweden) and normalised to standard temperature and pressure (STP 0 °C, 1 bar). pH monitoring used pH probes and IXIAN transmitters from Atlas scientific, calibrated at pH 4, 7, and 10 before each of the experimental periods. The pressure in the hydrogen injection line was measured(PXM 309 0.035GI, Omega, Norwalk, USA) to monitor pressure at the sparger and for safety shutoff purposes (>300 mBar). The ambient hydrogen concentration was monitored with a sensor (4H2-40000, Gasman, Devon, UK), which triggered an audible alarm (200 ppm) and safety shutoff (>1000 ppm).

### 2.6. In-Situ Biomethanation—Hydrogen Injection Control

For the biomethanation experiments, the estimated hydrogen injection requirement ($G_{H2\_est}$) was calculated as a function of the measured specific $CO_2$ yield in the relevant control reactor ($Y_{CO2}$), the measured OLR ($\dot{m}_{OLR}$), and the stoichiometric requirement (2) and set at the beginning of each experimental period. Considering that adding hydrogen on a stoichiometric basis for complete biomethanation can lead to buffer depletion and pH instability (as noted by [9]), 90% of the stoichiometric requirement was used:

$$G_{H2\_est} = 0.9 \times 4 \times Y_{CO2}\dot{m}_{OLR} \times \frac{1000}{60 \times 24} \tag{2}$$

where $G_{H2\_est}$ is expressed in the MFC rating units (mL min$^{-1}$). The hydrogen injection rate passed to the MFC ($G_{H2\_MFC}$) was then capped using a gain-scheduling approach as originally proposed by Bensmann et al. [10] using a series of scheduling equations based on headspace gas compositions and calculated as per Equation (3). In this work, pH was also used as a scheduling parameter, for which the constraint was set below the maximum

operational pH for food waste AD that was reported during biomethanation by [9]. The scheduling equations, setpoints/constraints, and gain parameters used are shown in Table 1 (chosen from earlier testing, documented in [29]).

$$G_{H2\_MFC} = MIN(G_{H2\_est}, G_{H2\_CH4}, G_{H2\_CO2}, G_{H2\_H2}, G_{H2\_pH}) \tag{3}$$

**Table 1.** Gain scheduling parameters and equations.

| Scheduling Parameter (Measured) | Unit | Setpoint Type | Setpoint Name and Value | Gain Parameter and Value | Calculated Hydrogen Injection Value (mL min$^{-1}$) and Scheduling Equation |
|---|---|---|---|---|---|
| CH$_4$ conc. ($\chi_{CH4}$) | [% vol]. | Setpoint | $S_{CH4\_sp}$= 90 | $k_{CH4}$= 0.3 | $G_{H2\_CH4} = k_{CH4}(S_{CH4\_sp} - \chi_{CH4})$ |
| CO$_2$ conc. ($\chi_{CO2}$) | [% vol.] | Min | $S_{CO2\_min}$= 5 | $k_{CO2}$= 0.3 | $G_{H2\_CO2} = k_{CO2}(\chi_{CO2} - S_{CO2\_min})$ |
| H$_2$ conc. ($\chi_{H2}$) | [% vol.] | Max | $S_{H2\_max}$= 40 | $k_{H2}$= 0.3 | $G_{H2\_H2} = k_{H2}(S_{H2\_max} - \chi_{H2})$ |
| pH | [-] | Max | $S_{ph\_max}$= 8.2 | $k_{pH}$= 5.0 | $G_{H2\_pH} = k_{pH}(S_{pH\_max} - pH)$ |

If any condition resulted in a negative hydrogen injection value (or below the minimum range of the MFC), then the value passed to the MFC was simply set to zero for that period. The actual hydrogen injection setpoint was updated for each cycle of the online GC.

*2.7. In Situ Biomethanation—Experimental Design*

For each feedstock (SS, FW), operation with biomethanation (i.e., hydrogen injection) was performed in duplicate (SS1, SS2 and FW1, FW2), alongside a single control reactor with no hydrogen injection (Control SS and FW), as summarised in Table 2.

**Table 2.** Reactor conditions during biomethanation experiments.

| Reactor number | 1 | 2 | 3 | 4 | 5 | 6 |
|---|---|---|---|---|---|---|
| Reactor name | Control SS | SS1 | SS2 | Control FW | FW1 | FW2 |
| Feedstock | Sewage Sludge (SS) | | | Food Waste (FW) | | |
| Biomethanation | No | Yes | | No | Yes | |

The experiment was split into two main stages (R1–R3 and O1–O3), which are summarised in Table 3. In the first stage (R1–R3), the study was focused on the effect of the recirculation rate on in-situ biomethanation. In this stage, the gas recirculation was varied into three different recirculation rates at 20, 120, and 280 rpm or the equivalent to 12, 67, and 155 L L$^{-1}$ day$^{-1}$. In the second stage (O1–O3), the process optimisation was carried out by modifying an operational condition, including an additional sparger on the recirculation line, a higher liquid mixing rate, and operating in a lower OLR, all the while maintaining the biogas recycling rate at the median value tested in the previous stage.

Liquid samples were taken twice a week to analyse the biological process indicators, such as TS, VS, alkalinity, ammonia, and volatile fatty acids. Prior to each period, all reactors were operated without hydrogen injection to establish the baseline biogas production and composition for at least 1 week or until stable biogas production was obtained.

*2.8. In Situ Biomethanation—Calculated Parameters*

The following are the calculated parameters that were used to characterise performance during experiments.

The gas retention time ($RT_G$) is an important process parameter which directly influences hydrogen conversion. The longer the hydrogen is in contact with the liquid phase, the higher the amount that will be dissolved and finally converted to methane. It is the

ratio between the overall system gas headspace ($V_H$) and total biogas outflow ($Q_{biogas}$), expressed by:

$$RT_G = \frac{V_H}{Q_{biogas}} \tag{4}$$

**Table 3.** Summary of experimental periods.

| Experimental Period | R1 | R2 | R3 | O1 | O2 | O3 |
|---|---|---|---|---|---|---|
| Variable of interest and short description | Recirculation flow rate | | | As R2 with sparger on recirculation line | As O1 with increased mixing rate | As O2 with reduced OLR |
| SS trial length (days) | 17 | 12 | 21 | 19 | 20 | 18 |
| FW trial length (days) | 14 | 9 | 21 | 32 | 20 | 18 |
| Biogas recirc. rate (RPM) | 20 | 120 | 240 | 120 | 120 | 120 |
| Biogas recirc. rate (L L$^{-1}$d$^{-1}$) | 12 | 67 | 115 | 67 | 67 | 67 |
| With/without 10μm sparger on recirculation line | No | | | Yes | Yes | Yes |
| Mixing stirring rate (RPM) | 60 | | | 60 | 110 | 110 |
| OLR (g VS L$^{-1}$ d$^{-1}$) | 2 | | | 2 | 2 | 1 |
| SS HRT (days) | 14 | | | 14 | 14 | 28 |
| FW HRT (days) | 68 | | | 68 | 68 | 136 |

Hydrogen conversion describes how much of the hydrogen injected is actually converted in the biomethanation process and is defined as:

$$X_{H2} = \frac{\sum Q_{H2,inj} - \sum Q_{H2,out} + \Delta H_2}{\sum Q_{H2,inj}} \tag{5}$$

taken over the various experimental periods, where $\sum Q_{H2,inj}$ is the sum of the hydrogen injected into the reactor and $\sum Q_{H2,out}$ is the sum of the hydrogen leaving the reactor, while $\Delta H_2$ is the volumetric variation in the H$_2$ amount contained in the headspace in a given time interval.

The methane evolution rate (MER) expresses the increase in the volumetric methane production rate, over the methane production from the background AD process, resulting from the in-situ biomethanation process and can be calculated using:

$$\text{MER} = \frac{\sum Q_{H2,inj} - \sum Q_{H2,out} + \Delta H_2}{4\,V_R} \tag{6}$$

The biomethanation extent expresses the extent to which biomethanation has increased the ratio of methane to carbon dioxide in the biogas and ignores any hydrogen content. It can be calculated using Equation (7), where $\varnothing$ represents the volume or molar fraction:

$$\text{Biomethanation extent} = \frac{\chi_{CH4}}{\chi_{CH4} + \chi_{CO2}} \tag{7}$$

The volumetric gas–liquid mass transfer coefficient ($k_L a$) is expressed in Equation (8) for hydrogen mass transfer, in a molar form, derived from the two-film theory [5,30]:

$$\dot{n}_{G/L} = V_R\,k_L a\left(C^*_{H2,l} - C_{H2,l}\right) \tag{8}$$

where $\dot{n}_{G/L}$ is the molar transfer rate (mol d$^{-1}$), $V_R$ is the reactor working volume, $C^*_{H2,l}$ and $C_{H2,l}$ are the (molar) equilibrium and actual liquid concentrations, respectively, and $k_L a$ is the gas–liquid transfer coefficient (d$^{-1}$).

Due to difficulties surrounding the measurement of $C_{H2,l}$, and the fact that in a healthy biogas system, the concentration of dissolved hydrogen is maintained at a very low value by the action of hydrogenotrophic methanogens, it is common to assume that the $C_l$ value is

negligible compared to $C_l^*$, e.g., [5]. This is equivalent to the assumption that the gas–liquid transfer rate of $H_2$ is the limiting step of the biomethanation reaction.

$C_{H2,l}^*$ can be evaluated knowing the concentration of the gas in the bulk phase $C_{H2,g}$ using Henry's Law:

$$C_{H2,l}^* = \frac{C_{H2,g}}{H} \tag{9}$$

where $H$ is the dimensionless Henry's constant; for hydrogen and water at 35 °C, the value is 50 [5].

Since the concentration of hydrogen, $C_{H2,g}$, changes spatially in the system (e.g., injected as pure hydrogen and mixed with other gasses in the headspace), a mean logarithmic concentration, shown in Equation (10), was used as the $C_{H2,g}$, as in [30]:

$$C_{H2,g} = \frac{C_{H2,\,g,injection} - C_{H2,\,g,headspace}}{\ln\left(C_{H2,\,g,injection}\right) - \ln\left(C_{g,headspace}\right)} \tag{10}$$

Combining the previous equations, it is possible to calculate the value of $k_L a$ for a given set of experimental measurements:

$$k_L a = \frac{H\,\dot{n}_{G/L}}{V_R\,C_{H2,g}} = \frac{H\left(\sum Q_{H2,inj} - \sum Q_{H2,out} + \Delta H_2\right)}{V_R\,C_{H2,g}}\left(\frac{p}{RT}\right) \tag{11}$$

where the molar flow, $\dot{n}_{G/L}$, is transformed into the measured volumetric flow (L d$^{-1}$) at STP conditions (temperature (T) 273.15 K, pressure (p) 1 bar, and gas constant R 0.08314 L bar K$^{-1}$ mol$^{-1}$).

*2.9. Mathematical Modelling*

A mathematical model was produced to better understand the interaction between various operational conditions and observed experimental results. An overview of the material fluxes considered is shown in Figure 3. The model was not designed to fully describe the biomethanation process but instead to support hypotheses relating to underlying mechanisms. The model is based on a single AD reaction that produces only methane and carbon dioxide:

$$\text{Feedstock} \rightarrow CH_{4,\,AD} + CO_{2,\,AD} + \text{Digestate} \tag{12}$$

The biomethanation follows Equation (1) and relies on a number of simplifying assumptions:

1. The background AD process (the degradation of biomass into biogas) occurs at a constant rate and fixed stoichiometry.

2. The $CO_2$ produced by the background AD process is dissolved and available for the biomethanation of injected hydrogen.

3. The reactor has fixed liquid and headspace volumes (i.e., the fedfeedstock volume is equal to the removed digestate volume) and the headspace acts as a fully mixed reactor compartment containing the gasses produced by the AD and biomethanation processed.

4. As per Section 2.8, the liquid concentration of dissolved hydrogen is negligible and therefore the process is mass transfer limited.

For ease of comparison with experimental data, a volumetric balance was made around the digester headspace, assuming all gasses behave ideally, at a fixed temperature and pressure. The dissolved gases fluxes were calculated as their equivalent in volume for convenience, although they themselves occupied no volume.

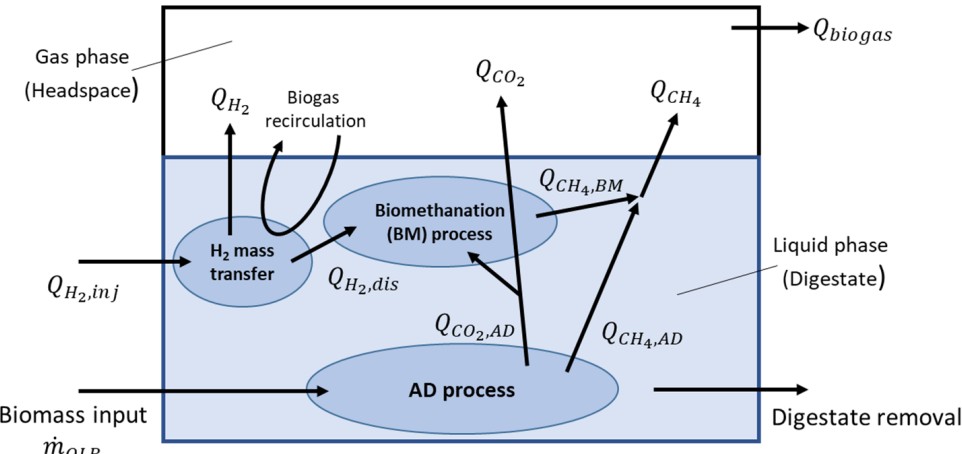

**Figure 3.** Overview of the mathematical model of the in-situ biomethanation process.

The background volumetric production of methane $(Q_{CH_4,AD})$ and carbon dioxide $(Q_{CO_2,AD})$ from the AD process are proportional to the OLR $(\dot{m}_{OLR})$ using:

$$Q_{CH_4,AD} = Y_{CH4}\dot{m}_{OLR}V_R \tag{13}$$

$$Q_{CO_2,AD} = Y_{CO2}\dot{m}_{OLR}V_R \tag{14}$$

where $k_{CH4}$ and $k_{CO2}$ are the specific methane and carbon dioxide production constants taken from the experimental data measured on the control reactors.

Based on the control system design as described in Section 2.6, the volumetric hydrogen injection, when it is not constrained by pH or gas composition, can be expressed by its stoichiometric requirement:

$$Q_{H_2,\,inj} = 0.9(4\,Q_{CO_2,AD}) \tag{15}$$

Based on Equation (8), the dissolved hydrogen flux, expressed volumetrically $(Q_{H_2,\,dis})$, at STP conditions, can be calculated using:

$$Q_{H_2,dis} = k_L a\left(\frac{RT}{p}\right)V_R C_L^* \tag{16}$$

where $C_L^*$ is calculated as in Equations (9) and (10).

The hydrogen $(Q_{H_2})$ and carbon dioxide $(Q_{CO_2})$ entering the headspace can be calculated by molar balance assuming that all the dissolved hydrogen reacts with carbon dioxide and that the unreacted carbon dioxide enters the headspace:

$$Q_{H_2} = Q_{H_2,inj} - Q_{H_2,dis} \tag{17}$$

$$Q_{CO_2} = Q_{CO_2,AD} - \frac{1}{4}Q_{H_2,dis} \tag{18}$$

Methane production from biomethanation $(Q_{CH_4,BM})$ is calculated based on reaction stoichiometry (19), and the total methane produced is the sum of that from AD and biomethanation (20), with the subsequent total biogas outflow $(Q_{biogas})$ being calculated using Equation (21):

$$Q_{CH_4,BM} = \frac{1}{4}Q_{H_2,\,dis} \tag{19}$$

$$Q_{CH_4} = Q_{CH_4,BM} + Q_{CH_4,AD} \tag{20}$$

$$Q_{biogas} = Q_{CH_4} + Q_{CO_2} + Q_{H_2} \tag{21}$$

The resulting equilibrium biogas composition can be calculated, based on a component volumetric balance by finding the steady state solution ($d\,\chi_{gas}/dt = 0$), where $\chi_{gas}$ is the volumetric composition in the headspace, to Equations (22)–(24) for all three component gasses ($CH_4$, $CO_2$, $H_2$), noting that the biogas recirculation is ignored during this balance since all mass transfer is accounted for in Equation (16) and that otherwise the recirculation does not result in a net transfer to or from the gas phase.

$$\frac{d\,\chi_{CH_4}}{t} = \frac{1}{V_H}\left(Q_{CH_4} - \chi_{CH_4}Q_{biogas}\right) \tag{22}$$

$$\frac{d\,\chi_{CO_2}}{t} = \frac{1}{V_H}\left(Q_{CO_2} - \chi_{CO_2}Q_{biogas}\right) \tag{23}$$

$$\frac{d\,\chi_{H_2}}{t} = \frac{1}{V_H}\left(Q_{H_2} - \chi_{H_2}Q_{biogas}\right) \tag{24}$$

where $V_H$ is the overall gas phase volume.

## 3. Results and Discussion

### 3.1. BMP and Baseline AD Process

The results of the characterisation of the inoculum and feedstock samples are presented in the Appendix A.1 (Tables A1 and A2). The BMP results are shown in Figure 4. The average methane production of the SS and FW during the BMP test was $0.402 \pm 0.005$ and $0.471 \pm 0.020$ L g$^{-1}$ VS, respectively. These are within the ranges given in the literature of 0.220–0.460 [31–33] and 0.460–0.530 [34–36] for SS and FW, respectively. Using the values of the elemental analysis for both feedstocks (Table A2) and the calculated theoretical methane potential [37], the BMP of SS and FW resulted in 76.04% and 87.95% of their respective theoretical potentials. The methane potential of the cellulose as a positive control using the two inoculums gave values of 0.370 and 0.374 L g$^{-1}$ VS, equivalent to 89.5% and 90.4% of the theoretical values.

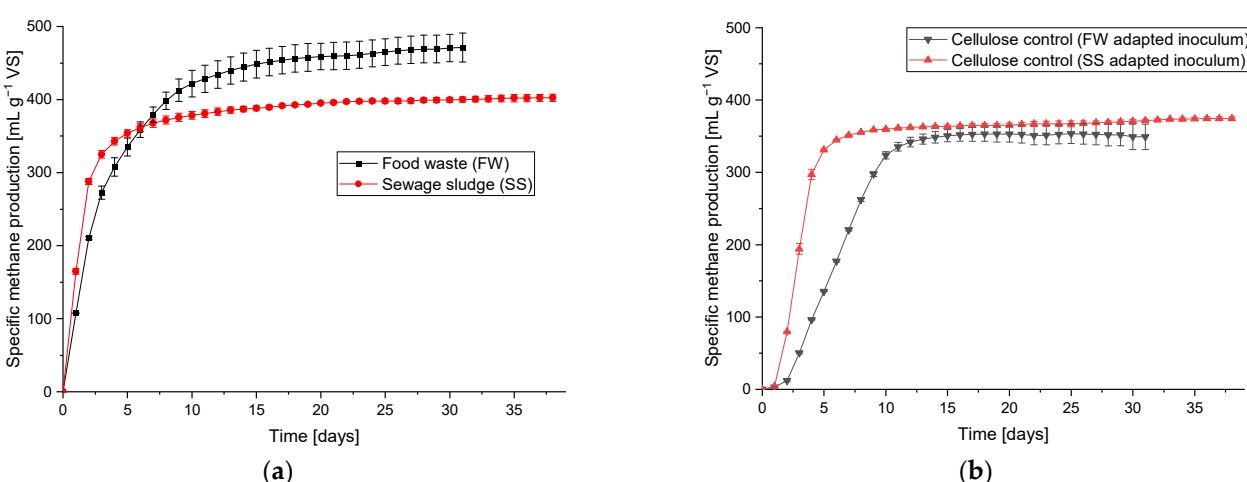

**Figure 4.** BMP test profiles for FW and SS (**a**), and the cellulose control for both inoculums (**b**).

Figure 5 shows the biogas composition from the initial baseline period prior to the biomethanation periods, which was similar in all replicates and validates the repeatability of the experimental approach; further baseline results are given in Appendix A.1 Table A3. The average methane and carbon dioxide specific yields were 0.24 and 0.12 L g$^{-1}$ VS for SS and 0.42 and 0.28 L g$^{-1}$ VS for FW, respectively. The baseline AD achieved 58 and 86% of the BMP specific methane yields, which is expected when comparing batch to continuous AD processes [38]. The lower result for SS can be attributed to its relatively short hydraulic retention time (HRT) (14 days, at OLR 2 g$^{-1}$ VS L$^{-1}$ day$^{-1}$) compared with the FW reactors (~68 days), since, in general, a longer HRT can be associated with a greater

degree of feedstock degradation, as shown for FW in [36]. During the baseline AD testing, the alkalinity ratio (IA/PA) for all reactors was stable, with it being approximately 0.40 (SS) and 0.38 (FW).

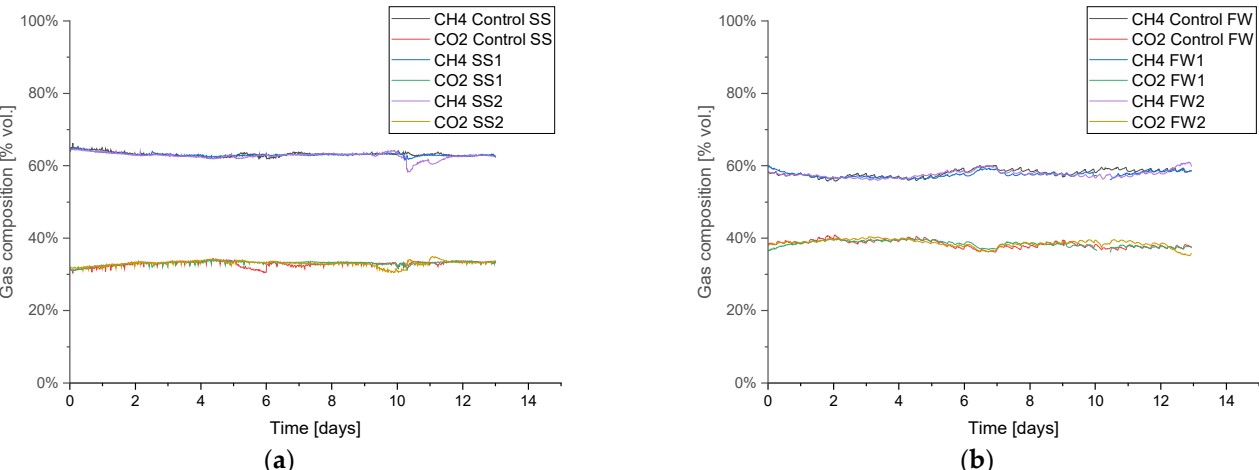

**Figure 5.** Biogas composition profile during a baseline AD testing prior to hydrogen injection, for sewage sludge (SS) (**a**) and food waste (FW) (**b**).

### 3.2. In-Situ Biomethanation Testing

For each experimental period (R1–R3, O1–O3), prior to hydrogen injection, it was ensured that all replicates had a similar baseline in terms of gas composition, gas content, pH, and alkalinity ratio. The hydrogen was then injected into the four biomethanation reactors (FW11, FW2, SS1, and SS2) in accordance with the specific experimental design of each period. The estimated hydrogen injection stoichiometric requirement ($G_{H2\_est}$) was updated before each experimental period based on the carbon dioxide production of the control prior to the start of hydrogen injection. The following sections will present the results for each of these periods, focusing on the results during hydrogen injection.

The volume of data collected, the amount of possible detailed discussion in terms of describing results, and the operational challenges would make a comprehensive discussion of all the experimental data too lengthy. Instead, a single experimental period (R1) and feedstock (SS) will be described in detail to exemplify the data gathered as well as the challenges and complexities faced. Subsequently, average data taken over the whole experimental periods will be used to assess the overall trends and subsequent implications. The average data also include the transient period at the initial hydrogen injection, which generally lasts up to two days; therefore, the average data can be considered a good approximation of the steady state performance of the various experimental conditions. The full dataset, containing the detailed experimental outputs for the six experimental periods for the all the replicates and controls, is available on a separate data repository [39], and for a more detailed treatment of individual experimental conditions the reader is directed to [29].

#### 3.2.1. In Situ Biomethanation Dynamics (Example Period R1 with SS)

Period R1 (gas recirculation 12 L L$^{-1}$day$^{-1}$) was monitored for 17 days, and the average measured OLR during the experiment for control SS, SS1, and SS2 were 1.91, 1.88 and 1.92 g VS. L$^{-1}$ day$^{-1}$, respectively. Hydrogen injection remained stable at its setpoint value (1.02 mL min$^{-1}$, equivalent to approximately 0.86 L L$^{-1}$ day$^{-1}$), excluding short periods of activation regarding the hydrogen gain loop (i.e., $G_{H2\_H2}$) of the feedback control on days 0 and 9 and a technical problem with the hydrogen injection system on day 6. The biogas output and OLR data for this period are shown in Figure 6.

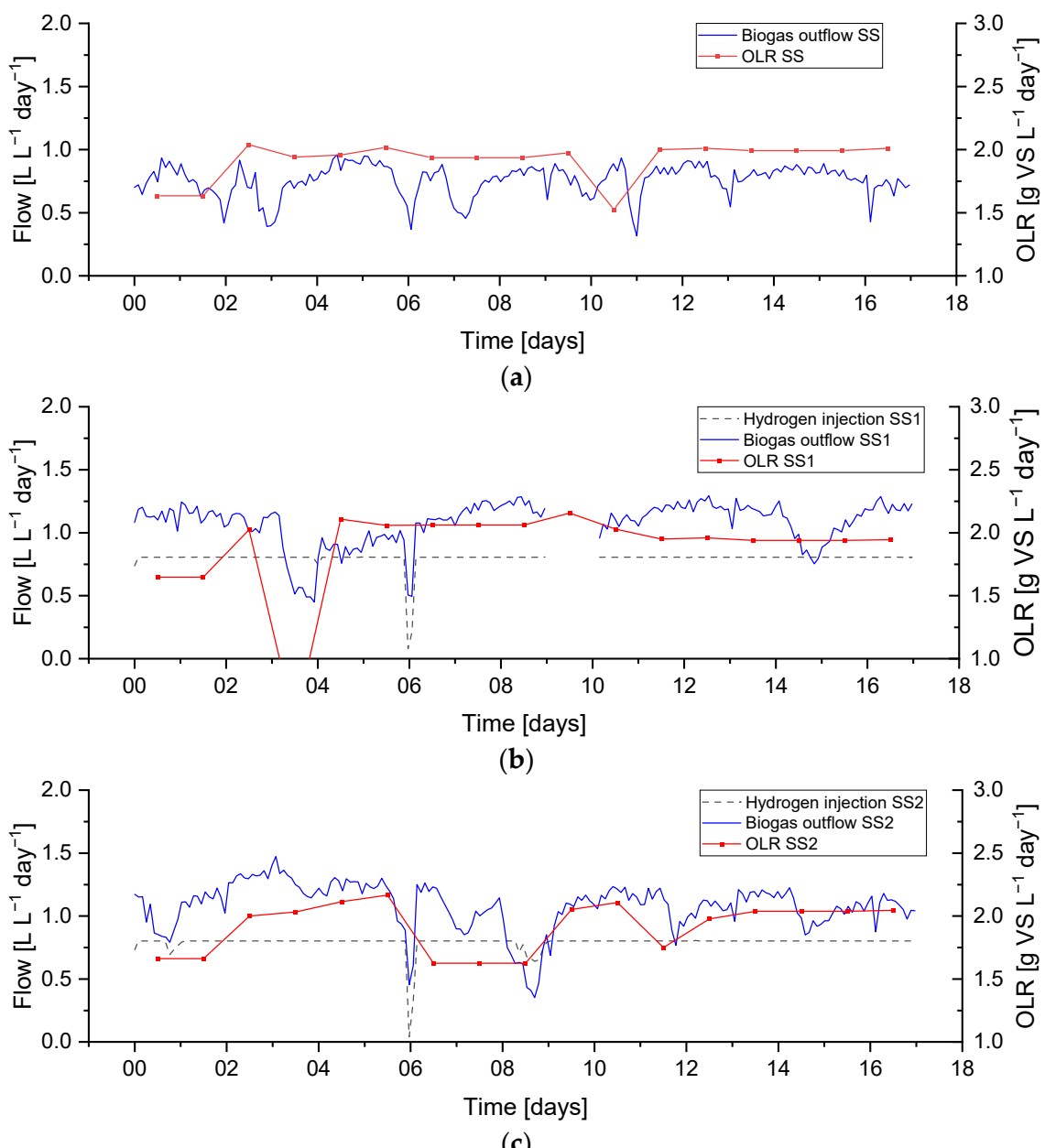

**Figure 6.** Biogas outflow, OLR, and hydrogen injection rate during period R1 for sewage sludge (SS)-fed reactors: (**a**) Control SS, (**b**) biomethanation reactor SS1, (**c**) biomethanation reactor SS2.

The initial pH, in all reactors, at the beginning of the experiment was around 7.1 (Figure 7). Indicative of the initiation of hydrogen injection in all experimental periods, the pH in the reactor SS1 and SS2 increased immediately after the addition of hydrogen, which was expected and previously reported due to bicarbonate consumption by the biomethanation reaction and the resulting predominance of the ammonia buffer on the carbonate buffer system [3]. Both duplicates showed similar pH profiles, with the average values for SS1 and SS2 being 7.41 and 7.44, respectively, compared with 7.20 in the case of the control reactor.

The duplicate reactors showed a similar biomethanation extent, which reached a value of approximately 80%, with the control value being approximately 70% (Figure 8), while the $H_2$ conversions were between 50–70%, with an average of 64 and 59% for SS1 and SS2, respectively. The $CO_2$ content decreased to approximately 15% in both biomethanation reactors and a 30% level in the control reactor. The methane also decreased in the biometha-

nation reactors, reaching an average of 56 and 54% in SS1 and SS2, respectively, due to the dilution by hydrogen in the headspace, which reached a maximum content of 33 and 36%, respectively.

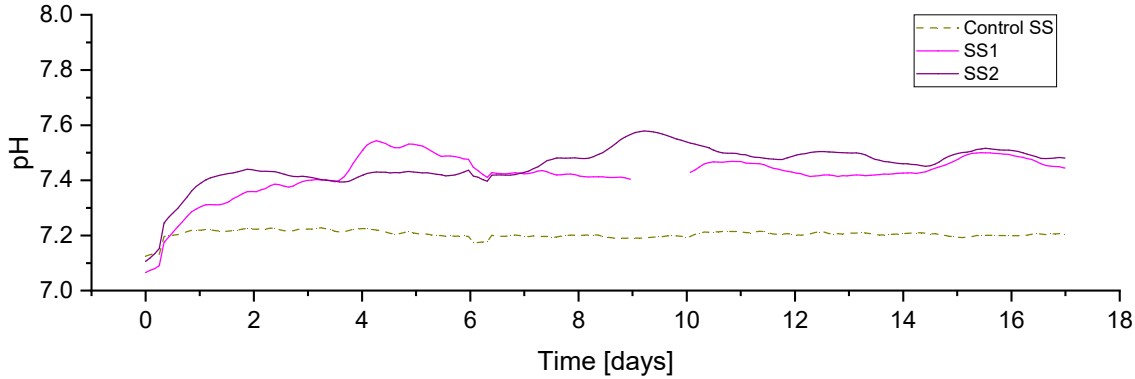

**Figure 7.** pH profile during period R1 for sewage sludge (SS) fed reactors.

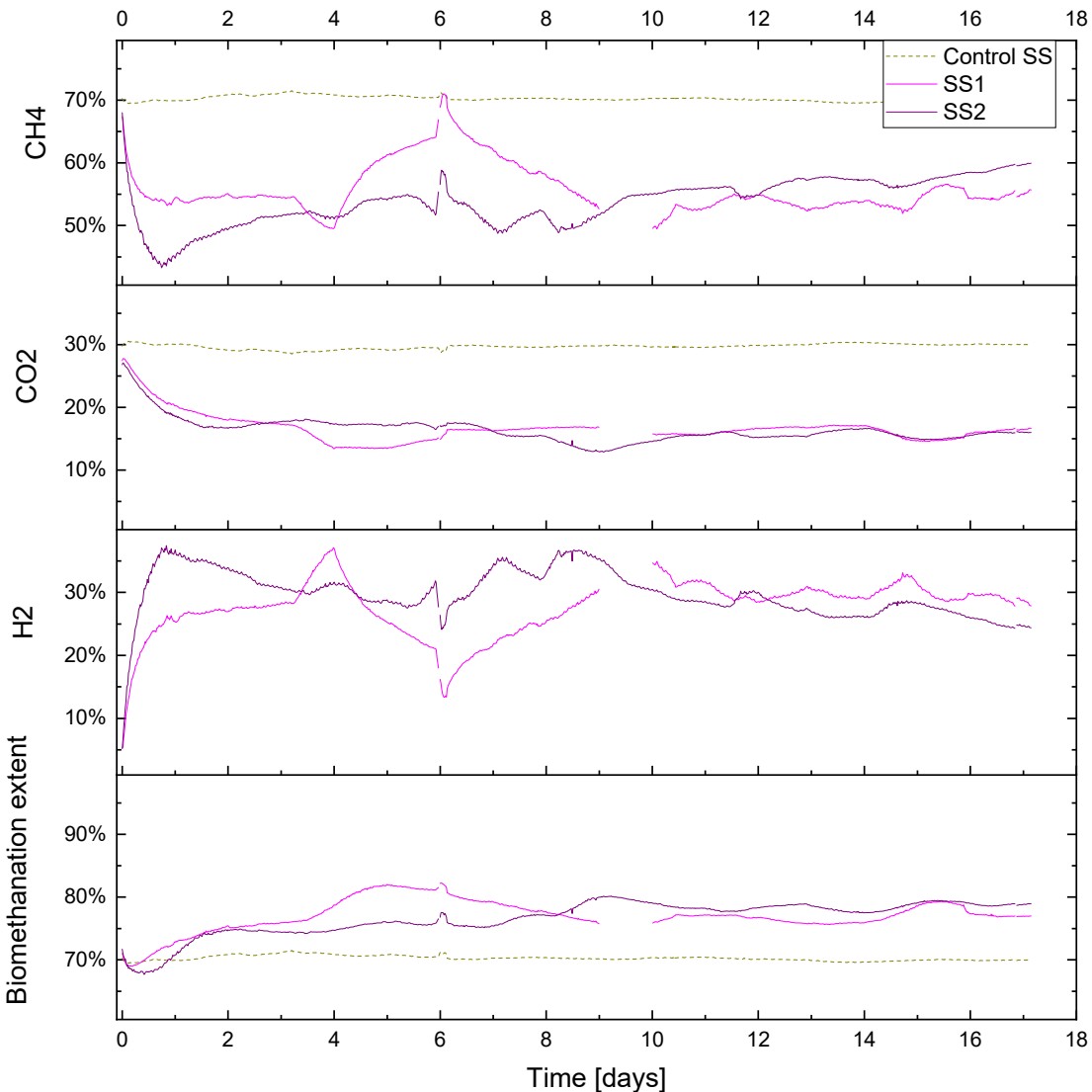

**Figure 8.** Gas composition ($CH_4$, $CO_2$, $H_2$ % vol.) and biomethanation extent (%) during period R1 for sewage sludge (SS) fed reactors.

A feedstock feeding pump failure on day 3 on reactor SS1 (the effect on the OLR can be seen in Figure 6) caused the feeding not to be delivered properly. This feeding failure caused an evident drop in biogas production, while the flow of hydrogen injection remained at its constant setpoint value. The lower biogas production and constant hydrogen injection caused an increase in the hydrogen concentration of up to 37%, while the methane and carbon dioxide concentrations dropped to 50 and 14%, respectively. In addition, the reduction in biogas production resulted in a higher retention time, leading to higher $H_2$ conversion, evident between days 3–4 in Figure 9.

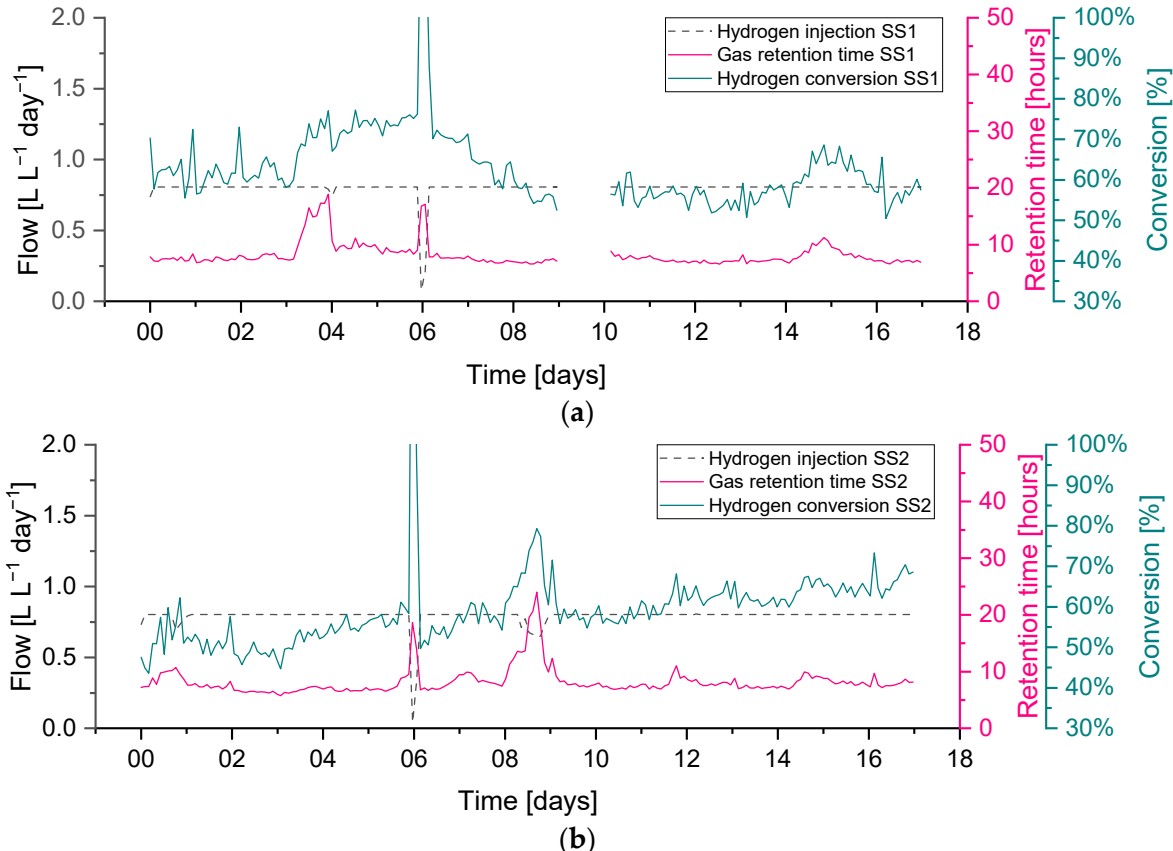

**Figure 9.** Hydrogen conversion in relation to gas retention time and hydrogen injection rate during period R1 for sewage sludge (SS)-fed reactors: (**a**) biomethanation reactor SS1, (**b**) biomethanation reactor SS2.

Abrupt changes in hydrogen injection also influence the process performance; this can be seen on day 6 in both reactors SS1 and SS2 when hydrogen had to be stopped for approximately five hours due to a technical issue. Biogas outflow immediately follows the change in hydrogen injection, and a similar reduction can be seen in Figure 6. This leads to a reduction in the gas flow and an increase in the gas residence time RT, which leads to a higher hydrogen conversion. The resulting changes in gas composition can be seen in Figure 8, where the hydrogen concentration diminishes and methane increase towards the value of the baseline; the carbon dioxide, on the other hand, has a slower response, and its content in the headspace remains more stable due to its increased solubility and residual buffering capacity in the liquid phase.

The hydrogen gain loop (i.e., $G_{H2\_H2}$) of the feedback controller was activated on days 0 and 9 on the reactor SS2, as shown as a small drop in the hydrogen injection flow (see Figure 6). The gas composition of methane and carbon dioxide remained far from the constraints, while the hydrogen was near the upper constraint (40%).

The hydrogen conversion tended to increase from 50% at the beginning to 70% at the end of the experiment in both reactors (Figure 9). This trend is also confirmed by the gas

composition data (Figure 8), where on the final six days of the experimental period, the methane content was observed to slightly increase in reactor SS2 along with a decrease in the hydrogen content. The $H_2$ conversion trend can be explained by microbial acclimation and the growth of the hydrogenotrophic population during the experiment. Microbial activity can, in fact, increase the gas–liquid mass transfer rate compared to a purely physical process in abiotic liquid by converting the absorbed gas into the stagnant liquid layer surrounding the gas bubble, thereby increasing the diffusional gradient. This phenomenon is reported in the literature as the microbial enhancement of the gas–liquid mass transfer [30].

3.2.2. Variation in Biogas Recirculation Rate (Periods R1–R3)

A summary of the average values of the main process parameters across periods R1–R3 (i.e., the variation in the biogas recirculation rate) for both SS and FW can be found in Figure 10. These data are supplemented by more detailed (per reactor) results presented in the Appendix A.2 (Tables A4 and A6). During periods R1–R3, the general observations and trends are similar for the SS and FW fed reactors, despite the generally higher all gas flow values in the FW reactors due to the high BMP of FW compared with SS. Consequently, the discussion will focus on SS whilst highlighting deviations from these for the FW reactors.

The general trend is that the increasing recirculation rate improved the hydrogen consumption rate. On average, the hydrogen conversion increased from 0.84 to 1.09 L day$^{-1}$ between periods R1–R3 and increased the specific methane production rate by 26% compared with the control reactor. The hydrogen conversion rates constituted with the hydrogen injection rate in all periods were between 60 and 75%, where the highest value was achieved at R3–SS. This conversion rate was lower than that obtained previously [12] using a UASB reactor and a ceramic sponge diffuser (86.8%), but much larger than that obtained from a large-scale reactor (10–26%) [15].

Increasing the biogas recirculation rate also improves the methane evolution rate (MER) from 0.12 L L$^{-1}$ day$^{-1}$ to 0.15 L L$^{-1}$ day$^{-1}$. In general, the MER of in-situ biomethanation is in the range of 0.08 to 0.39 L L$^{-1}$ day$^{-1}$ [40].

An anomaly appears when analysing the consumption of hydrogen and the volume of methane that is produced additional to the control, which should theoretically be 0.25 L $CH_4$ per L of $H_2$ consumed. For example, in R1, the ratio of the additional methane enrichment with hydrogen converted was 0.15 L $CH_4$ L$^{-1}$ $H_2$. In R3, the ratio increased to 0.23 L $CH_4$ L$^{-1}$ $H_2$ (0.27 for FW). Beyond any undetected experimental error, an explanation for this could be that hydrogen is being consumed for microbial growth [5,17]; a $H_2$:$CH_4$ ratio above 4 has been suggested to account for microbial biomass growth [41]. Related to this, the sum of the theoretical consumption of $CO_2$ by biomethanation (calculated from converted $H_2$) and the volumetric $CO_2$ from the output gas in all periods was 35%–39% higher compared to the $CO_2$ produced in the reactor control. In this case, the extra $CO_2$ might have come from bicarbonate consumption and reduced final dissolved $CO_2$, as has been observed previously [12], or could be the increased biochemical production of carbon dioxide due to differing process conditions between the control reactor and biomethanation reactors (cf. average pH in Appendix A), meaning that the assumed parity between the background AD process is invalid. The experimental overestimation of hydrogen consumption could explain both of these observations but a thorough examination of all measurements and calculations did not yield an opportunity for such an error.

The average pH values on the SS1 and SS2 increased compared to the pH at the reactor control. The average pH tends to increase along with the increase in the recirculation rate, owing to the increase in hydrogen consumption and therefore reduced carbon dioxide concentration that buffers the digestate pH. The average alkalinity during the experiment in SS1 and SS2 was improved along with the increasing recirculation rates. The average alkalinity ratio (IA/PA) in R1, 2, and 3 were 0.41, 0.35, and 0.28, respectively, with the recommended threshold of ratio being 0.3 [26]. The average total VFA equally decreased from an average of 2 to 1.2 g L$^{-1}$.

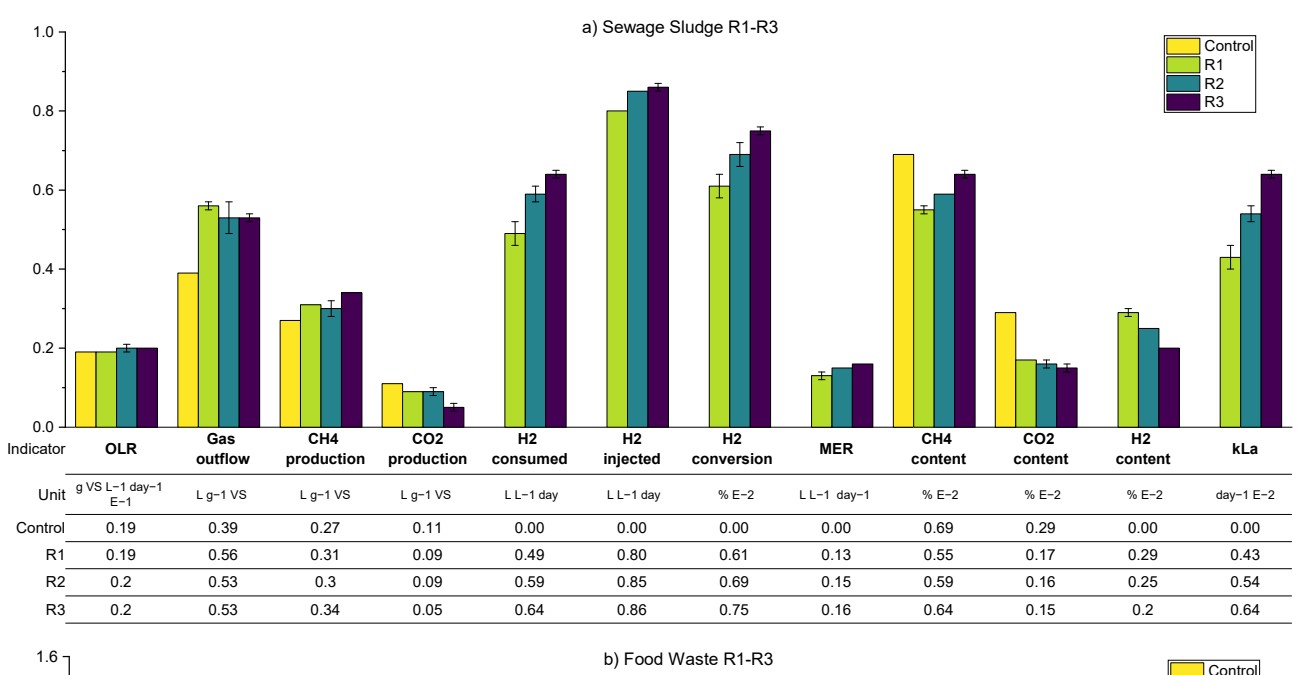

| Indicator | OLR | Gas outflow | CH4 production | CO2 production | H2 consumed | H2 injected | H2 conversion | MER | CH4 content | CO2 content | H2 content | kLa |
|---|---|---|---|---|---|---|---|---|---|---|---|---|
| Unit | g VS L−1 day−1 E−1 | L g−1 VS | L g−1 VS | L g−1 VS | L L−1 day | L L−1 day | % E−2 | L L−1 day−1 | % E−2 | % E−2 | % E−2 | day−1 E−2 |
| Control | 0.19 | 0.39 | 0.27 | 0.11 | 0.00 | 0.00 | 0.00 | 0.00 | 0.69 | 0.29 | 0.00 | 0.00 |
| R1 | 0.19 | 0.56 | 0.31 | 0.09 | 0.49 | 0.80 | 0.61 | 0.13 | 0.55 | 0.17 | 0.29 | 0.43 |
| R2 | 0.2 | 0.53 | 0.3 | 0.09 | 0.59 | 0.85 | 0.69 | 0.15 | 0.59 | 0.16 | 0.25 | 0.54 |
| R3 | 0.2 | 0.53 | 0.34 | 0.05 | 0.64 | 0.86 | 0.75 | 0.16 | 0.64 | 0.15 | 0.2 | 0.64 |

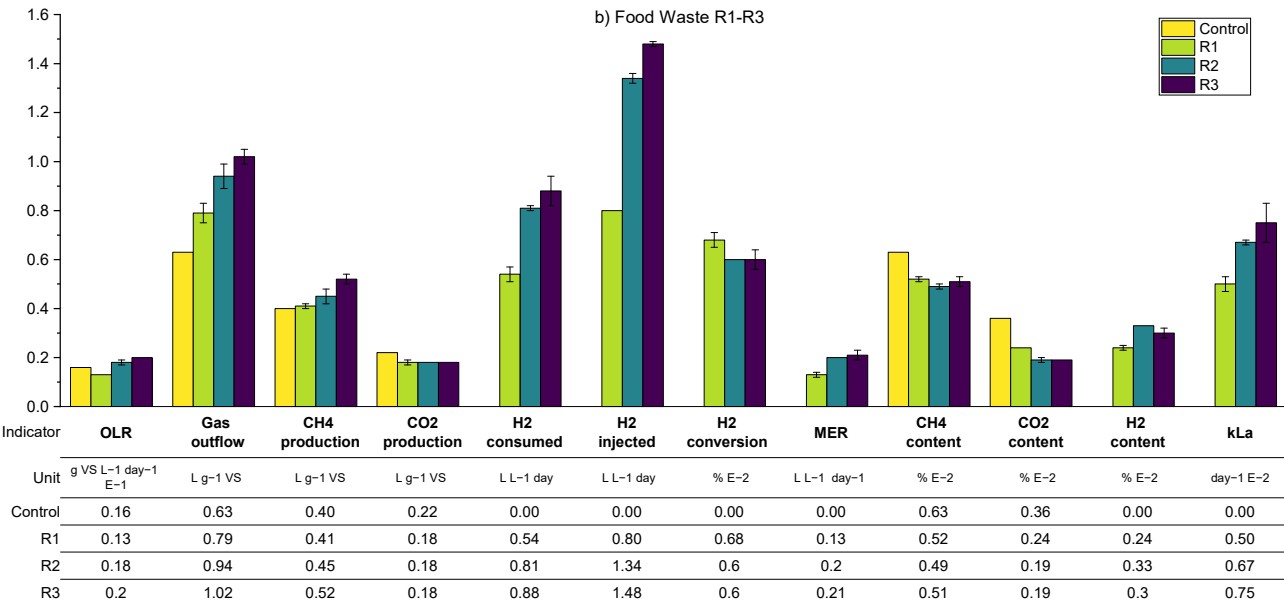

| Indicator | OLR | Gas outflow | CH4 production | CO2 production | H2 consumed | H2 injected | H2 conversion | MER | CH4 content | CO2 content | H2 content | kLa |
|---|---|---|---|---|---|---|---|---|---|---|---|---|
| Unit | g VS L−1 day−1 E−1 | L g−1 VS | L g−1 VS | L g−1 VS | L L−1 day | L L−1 day | % E−2 | L L−1 day−1 | % E−2 | % E−2 | % E−2 | day−1 E−2 |
| Control | 0.16 | 0.63 | 0.40 | 0.22 | 0.00 | 0.00 | 0.00 | 0.00 | 0.63 | 0.36 | 0.00 | 0.00 |
| R1 | 0.13 | 0.79 | 0.41 | 0.18 | 0.54 | 0.80 | 0.68 | 0.13 | 0.52 | 0.24 | 0.24 | 0.50 |
| R2 | 0.18 | 0.94 | 0.45 | 0.18 | 0.81 | 1.34 | 0.6 | 0.2 | 0.49 | 0.19 | 0.33 | 0.67 |
| R3 | 0.2 | 1.02 | 0.52 | 0.18 | 0.88 | 1.48 | 0.6 | 0.21 | 0.51 | 0.19 | 0.3 | 0.75 |

**Figure 10.** Summarised biomethanation experimental results; periods R1–R3 using SS (**a**) and FW (**b**). Error bars show duplicate reactor averages, and E notation is used to scale results to the same vertical axis.

As expected, the calculated $k_L a$ increased through the experimental periods due to the increased flow rate in the recirculation stream (greater gas holdup in the liquid phase and therefore higher specific area in terms of the bubbles). While this increase in mass transfer allowed for the significant consumption of hydrogen by the reactors, the presence of high concentrations of remaining hydrogen in the biogas outflow indicated that the process was mass transfer limited in all cases.

In the case of FW, despite similar trends, the hydrogen content was observed to be stable at a higher level compared to the equivalent SS reactors. The hydrogen gain loop was activated for the majority of the time during R1–R3, mainly caused by the larger hydrogen injection requirement of the feedstock, but with similar mass transfer characteristics. The methane concentration in the biogas outflow was lower for FW due to dilution with a higher amount of hydrogen but also the lower methane content from the background FW digestion.

To explore the variations in the gathered data over each operational period, in recognition of the broad variation in the measured data and performance parameters and the difficulty in terms of comparing the different operating conditions, the gas retention time ($RT_G$) was plotted against the hydrogen conversion for each biomethanation reactor and experimental period, as seen in Figure 11. Both variables were calculated as two hours average, covering the whole experimental period. Trendlines have been added of the form:

$$y = kx/(1 + kx) \qquad (25)$$

as suggested for a gas–liquid mass transfer limited process [42] since they offer a good representation of the observed trends, i.e., that the hydrogen conversion eventually appears to saturate with respect to an increased gas retention time ($RT_G$). The curves were fitted using OriginPro® to elucidate trends from the highly scattered data.

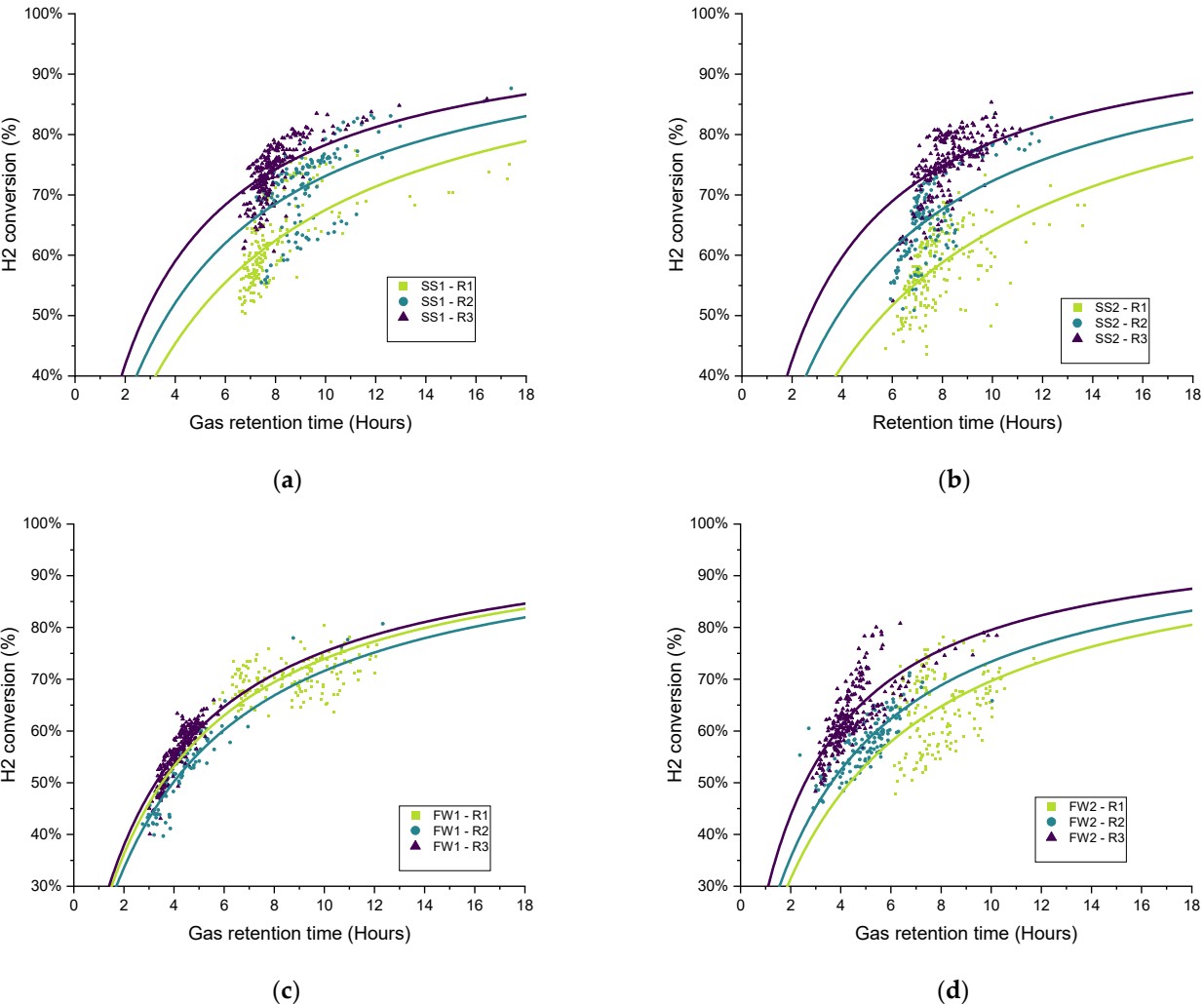

**Figure 11.** Scatter plot of the retention time and hydrogen conversion (two-hour period averages) with variations in biogas recirculation rate (R1–R3) for all biomethanation reactors: (**a**) SS1, (**b**) SS2, (**c**) FW1, and (**d**) FW2.

In general, the data distribution shows that hydrogen conversion increases along with the increase in the recirculation rate since the trendlines are also "ranked" in the graph following the same, from lowest to highest recirculation rates (R1–R3), with this result being previously reported [17]. There is a certain amount of deviation between the results of duplicates FW1 and FW2, which can also be seen in the hydrogen consumption and $k_La$ results for FW (Figure 10).

### 3.2.3. Further Optimisation of In-Situ Biomethanation (O1–O3)

As per the previous results, a summary of the average process parameters for periods O1–O3 for both SS and FW reactors are shown in Figure 12, with a more detailed breakdown in the Appendix A.2 (Tables A5 and A7). The expectation for the further optimisation periods was that all three interventions considered, additional sparger on the biogas recirculation line (O1), an increase in the mechanical mixing rate (O2), and a reduction in the OLR (O3), should improve the overall performance of the biomethanation process. The mechanism for this would be through increased mass transfer for the recirculation stream in O1 and both the injection and recirculation stream in O2, while for O3, the improved performance would result from the increased gas retention time due to generally lower biogas production and hydrogen injection.

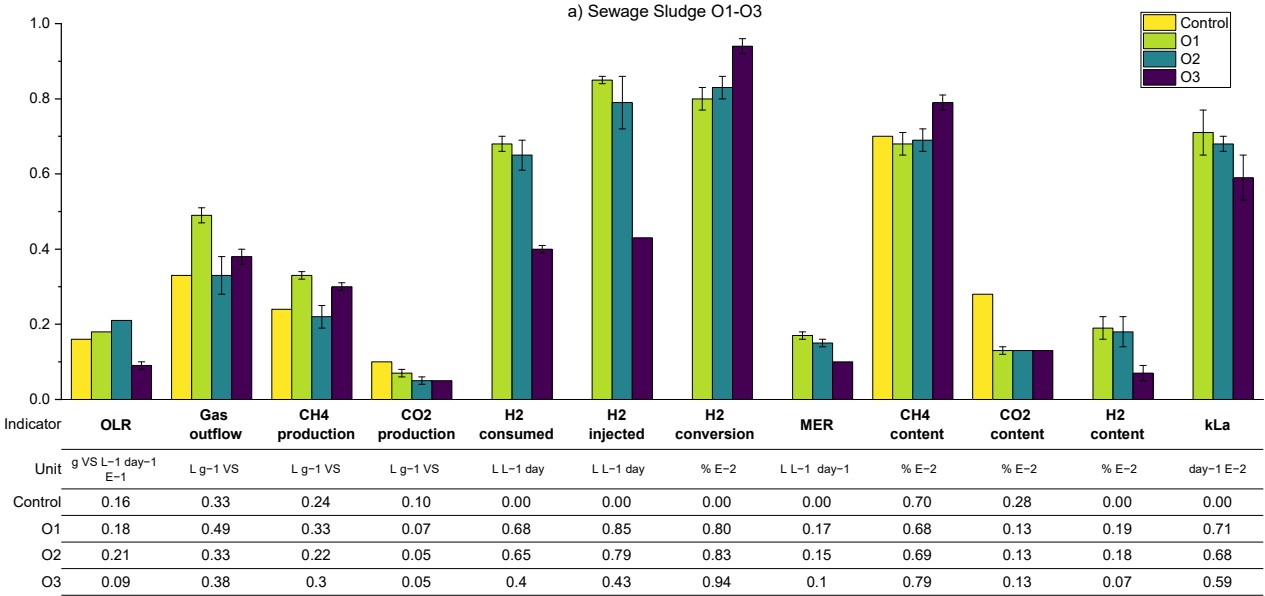

| Indicator | OLR | Gas outflow | CH4 production | CO2 production | H2 consumed | H2 injected | H2 conversion | MER | CH4 content | CO2 content | H2 content | kLa |
|---|---|---|---|---|---|---|---|---|---|---|---|---|
| Unit | g VS L−1 day−1 E−1 | L g−1 VS | L g−1 VS | L g−1 VS | L L−1 day | L L−1 day | % E−2 | L L−1 day−1 | % E−2 | % E−2 | % E−2 | day−1 E−2 |
| Control | 0.16 | 0.33 | 0.24 | 0.10 | 0.00 | 0.00 | 0.00 | 0.00 | 0.70 | 0.28 | 0.00 | 0.00 |
| O1 | 0.18 | 0.49 | 0.33 | 0.07 | 0.68 | 0.85 | 0.80 | 0.17 | 0.68 | 0.13 | 0.19 | 0.71 |
| O2 | 0.21 | 0.33 | 0.22 | 0.05 | 0.65 | 0.79 | 0.83 | 0.15 | 0.69 | 0.13 | 0.18 | 0.68 |
| O3 | 0.09 | 0.38 | 0.3 | 0.05 | 0.4 | 0.43 | 0.94 | 0.1 | 0.79 | 0.13 | 0.07 | 0.59 |

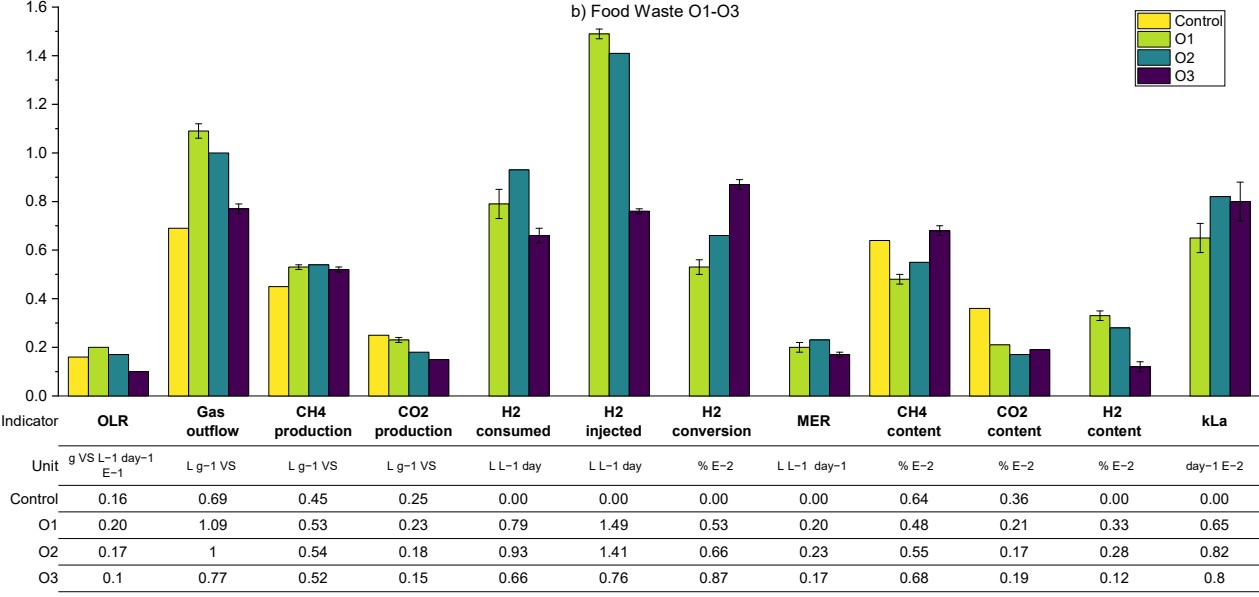

| Indicator | OLR | Gas outflow | CH4 production | CO2 production | H2 consumed | H2 injected | H2 conversion | MER | CH4 content | CO2 content | H2 content | kLa |
|---|---|---|---|---|---|---|---|---|---|---|---|---|
| Unit | g VS L−1 day−1 E−1 | L g−1 VS | L g−1 VS | L g−1 VS | L L−1 day | L L−1 day | % E−2 | L L−1 day−1 | % E−2 | % E−2 | % E−2 | day−1 E−2 |
| Control | 0.16 | 0.69 | 0.45 | 0.25 | 0.00 | 0.00 | 0.00 | 0.00 | 0.64 | 0.36 | 0.00 | 0.00 |
| O1 | 0.20 | 1.09 | 0.53 | 0.23 | 0.79 | 1.49 | 0.53 | 0.20 | 0.48 | 0.21 | 0.33 | 0.65 |
| O2 | 0.17 | 1 | 0.54 | 0.18 | 0.93 | 1.41 | 0.66 | 0.23 | 0.55 | 0.17 | 0.28 | 0.82 |
| O3 | 0.1 | 0.77 | 0.52 | 0.15 | 0.66 | 0.76 | 0.87 | 0.17 | 0.68 | 0.19 | 0.12 | 0.8 |

**Figure 12.** Summarised biomethanation experimental results; periods O1-O3 using SS (**a**) and FW (**b**). Error bars show duplicate reactor averages, and E notation is used to scale results to the same vertical axis.

For the SS reactors in period O1, this prediction was confirmed, by a comparison with the results from R2 (equivalent biogas recirculation rate), through an increase in $k_La$ to 71 day$^{-1}$ (c.f. 54) and hydrogen conversion to 80% (c.f. 60%). In period O2, however, the results show no improvement from the increase in the mechanical mixing rate, and even a slight, unexplained reduction in performance for SS2, except for the fact that the hydrogen conversion increased in O2 to 83% (c.f. 80% in O1). The periods are difficult to compare since there is a variation in the OLR from 1.8 to 2.0 gVS L$^{-1}$ day$^{-1}$ in O1 and O2, respectively. A higher OLR would lead towards shorted gas retention times and a reduction in hydrogen consumption, with other conditions remaining the same.

The best biomethanation performance in terms of desirable output biogas composition (high methane (79%), low carbon dioxide (13%), and hydrogen (7%)) and highest hydrogen conversion (94%) was achieved during O3 with a reduced in the OLR of 0.9 g VS L$^{-1}$ day$^{-1}$. This is despite a reduction in the observed $k_La$, which was expected due to the reduced hydrogen injection rate (in proportion with the reduced OLR) and therefore reduced gas hold-up. This increased performance is traded off against a lower production rate in terms of methane, expressed by a MER of 0.1 L L$^{-1}$ day$^{-1}$ (c.f. 0.15 in O2), and, due to the very low OLR used, this is unlikely to be a practical solution for improved biomethanation performance in a real-world scenario.

With regard to the FW experiments, O1–O3, a comparison between O1-FW (with the additional sparger) and R2-FW (with equivalent biogas recirculation rate) shows a slight reduction in hydrogen consumption, $k_La$, and the outflowing biogas composition. It is possible that the increased OLR in O1 (2.0 c.f. 1.8 g VS L$^{-1}$ day$^{-1}$ in R2) is masking any improvements in performance. It was also observed during O1 that there was an increase in the ammonia concentration (from 3.62 to 3.95 gTAN kg$^{-1}$) compared with previous experimental runs and, also, for the first time, foaming was detected. There are numerous causes for foaming in AD systems, such as improper mixing, fluctuations in the OLR, and substrate types [43]. The levels of VFA and alkalinity ratio in the biomethanation reactors were, on average, comparable to the levels in the control reactors, so no indication of biological instability was noted.

The addition of mechanical mixing in O2 did lead to improvements in biomethanation performance compared with R2 in terms of a higher hydrogen conversion (66 c.f. 60%), MER (0.23 cf. 0.20 L L$^{-1}$ day$^-$), and $k_La$ (82 c.f. 67 day$^{-1}$), with the trend being similar to that of O2 to R2 for the SS reactors. As per the SS reactors, the best biomethanation performance for FW in terms of hydrogen conversion and desirable output biogas composition was at the reduced OLR in O3.

The distribution of the hydrogen conversion as a function of the gas retention time for experimental periods O1–O3 for both SS and FW is shown in Figure 13. Similar to Figure 11, the fitted curves were added only for illustration rather than to imply quality of fit. On the whole, the trends observed and discussed above can be confirmed in the scatter plots, in that the biomethanation performance ranked in terms of hydrogen conversion for SS was R2 < O1 ≈ O2 < O3. For FW, O1 (additional sparger) did not see the process gains in one of the duplicates (FW1), and the ranking can be ordered as R2 ≈ O1 < O2 < O3 (as discussed above).

### 3.3. In-Situ Biomethanation Modelling

The model developed in Section 2.9 was used to simulate the conditions for the in-situ biomethanation experiments for two periods, O1 for SS and O2 for FW, as these configurations showed the overall best results in terms of biomethanation performance considering a combination of product quality (i.e., a high hydrogen consumption rate and a high/low concentration of methane/carbon dioxide in biogas) and quantity (the methane evolution rate).

Only minimal experimental data were required for the simulation owing to the low model complexity, i.e., no reaction kinetics, inhibition, and biochemical considerations. OLR and hydrogen injection were the model input. Only five parameters were required by

the model structure: the specific yields of methane and carbon dioxide produced by the background AD process ($Y_{CH4}$, $Y_{CO2}$) from the control reactor, the volumetric gas–liquid mass transfer coefficient ($k_L a$) from the biomethanation experimental data, and the reactor and headspace volumes ($V_R$, $V_H$).

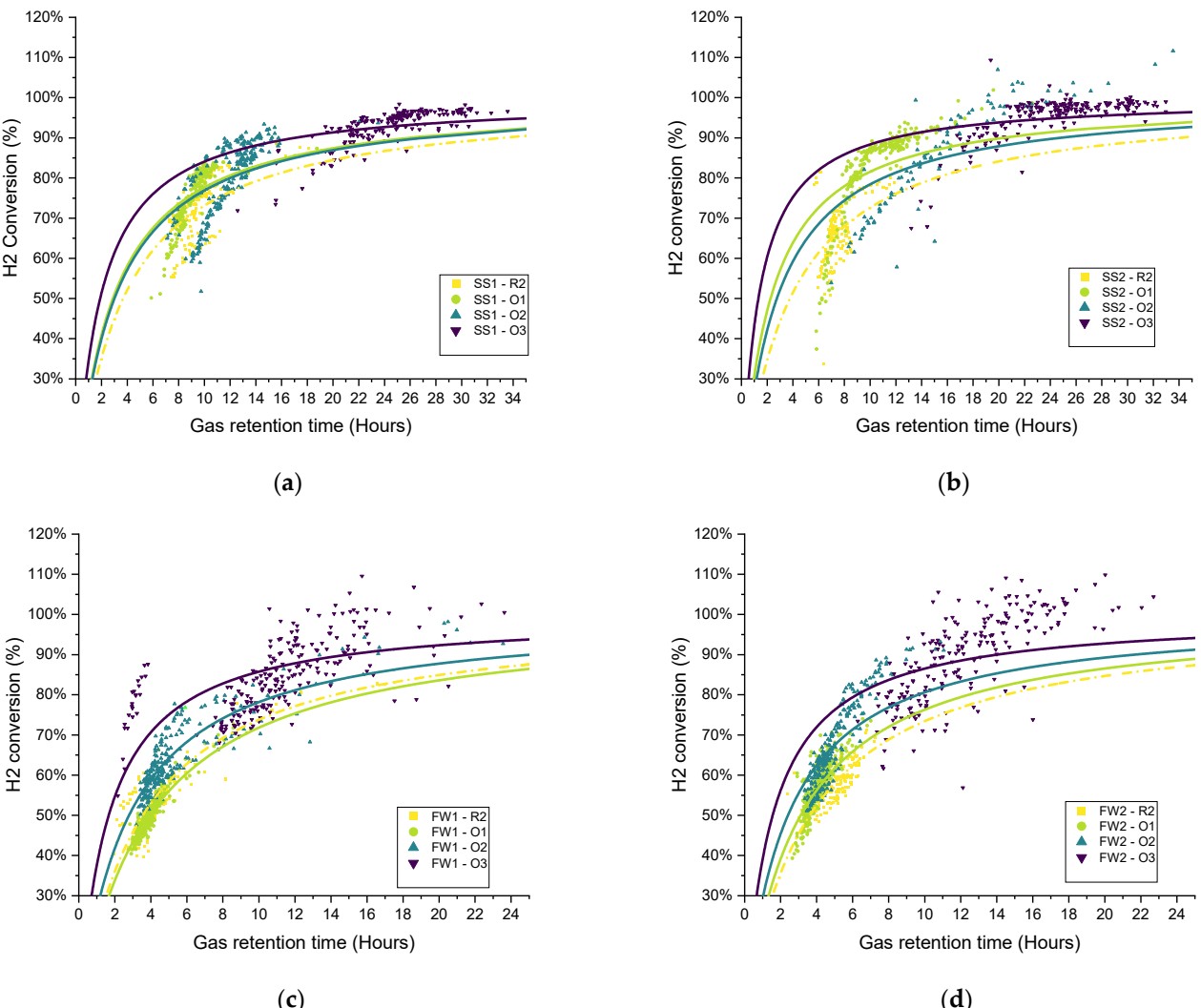

**Figure 13.** Scatter plot of the retention time and hydrogen conversion (two-hour periods average) with variations in operational conditions (O1–O3) for all biomethanation reactors:(**a**) SS1, (**b**) SS2, (**c**) FW1, and (**d**) FW2.

Despite a simplified process model, comparison between the simulated and experimental data show a good fit in terms of reproducing the main experimental average outputs, as shown in Figure 14. Apart from carbon dioxide specific yield and concentration in biogas, all the main results were fitted with a relative error below 16 and 11% for SS and FW, respectively. Across both feedstocks, carbon dioxide production and content were poorly represented, which can be related back to the discussion in 3.2.2 surrounding the mass balance between the observed hydrogen consumption and carbon dioxide production. Any mechanism that leads to increased observed carbon dioxide production (e.g., release through alkalinity/pH change or changes to the background AD process) was not included in the model, and therefore this experimental anomaly was not replicated.

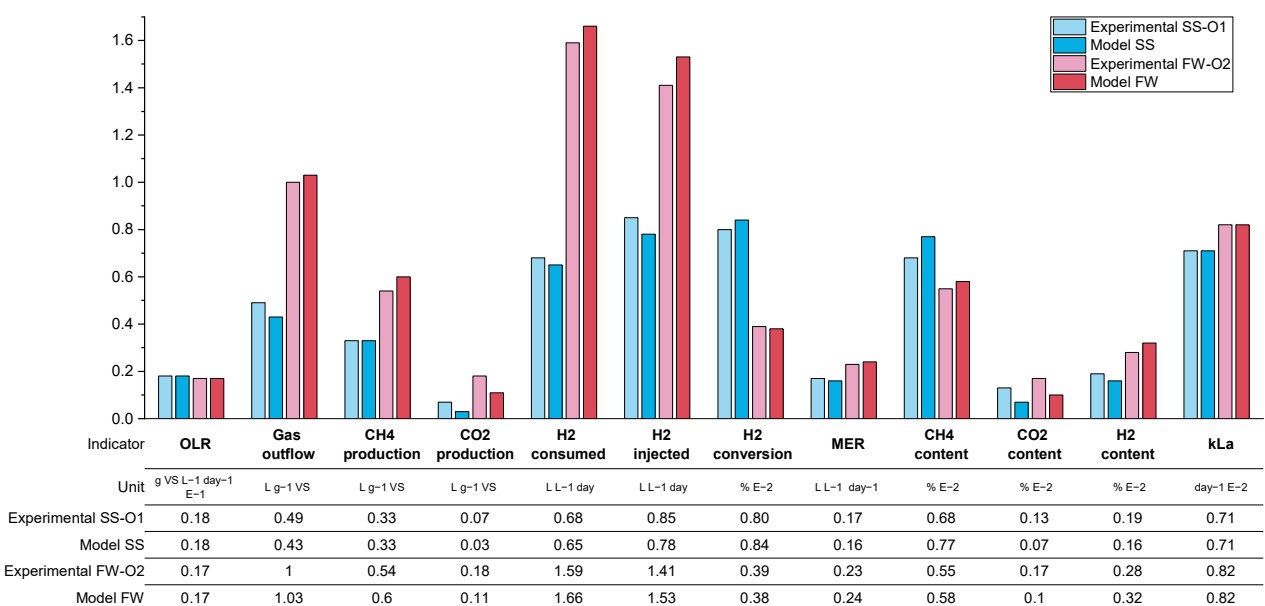

| Indicator | OLR | Gas outflow | CH4 production | CO2 production | H2 consumed | H2 injected | H2 conversion | MER | CH4 content | CO2 content | H2 content | kLa |
|---|---|---|---|---|---|---|---|---|---|---|---|---|
| Unit | g VS L−1 day−1 E−1 | L g−1 VS | L g−1 VS | L g−1 VS | L L−1 day | L L−1 day | % E−2 | L L−1 day−1 | % E−2 | % E−2 | % E−2 | day−1 E−2 |
| Experimental SS-O1 | 0.18 | 0.49 | 0.33 | 0.07 | 0.68 | 0.85 | 0.80 | 0.17 | 0.68 | 0.13 | 0.19 | 0.71 |
| Model SS | 0.18 | 0.43 | 0.33 | 0.03 | 0.65 | 0.78 | 0.84 | 0.16 | 0.77 | 0.07 | 0.16 | 0.71 |
| Experimental FW-O2 | 0.17 | 1 | 0.54 | 0.18 | 1.59 | 1.41 | 0.39 | 0.23 | 0.55 | 0.17 | 0.28 | 0.82 |
| Model FW | 0.17 | 1.03 | 0.6 | 0.11 | 1.66 | 1.53 | 0.38 | 0.24 | 0.58 | 0.1 | 0.32 | 0.82 |

**Figure 14.** Comparison of the experimental data and modelled data for SS period O1 and FW period O2.

The qualitative comparison between the modelling and experimental data also validates that the assumptions made during model formulation are also likely to be valid for the experiment. The background AD process appears to be relatively unaffected by the addition of additional hydrogen injection, and the process appears to be mass transfer limited.

The model can be used more broadly to explore the performance of an in-situ biomethanation system, as long as these founding assumptions hold. For demonstration, process contour maps were generated (Figure 15) showing the effect of variations in the OLR and hydrogen injection ratio, S (defined as $S = Q_{H_2,inj}/(4Q_{CO_2,AD})$), on a range of process characteristics and performance indicators, for three levels of mass-transfer: 60 day$^{-1}$, corresponding approximately to the conditions explored in this paper, and then 120 and 240 day$^{-1}$. The delivery of these improvements in mass transfer characteristics would require the redesign of the equipment, e.g., the use of membranes, increasing the aspect ratio (to increase the bubble path length), or a reduction in bubble size, but, given other works in this area, these values are not considered unrealistic.

Generally speaking, the model predicts that, within the explored conditions, in the best case, the process cannot deliver high performance single reactor in-situ biomethanation in a practical and sensible range in terms of the OLR commonly found in AD systems. For example, even at the highest $k_L a$ explored (240 day$^{-1}$), at an OLR of 5 g VS L$^{-1}$ day$^{-1}$ (Point A, Figure 15), the process is predicted to produce biogas upgraded to contain ~10% both carbon dioxide and hydrogen with ~80% methane. This gas would require further treatment for most current applications. For use as a biomethane (as a natural gas drop-in replacement), it is likely that some kind of $CO_2$ removal would be required, either through a further methanation step (e.g., ex situ biomethanation) or through a physical separation. For use as a vehicle fuel (as a CNG drop-in replacement), this gas would likely require hydrogen removal through, e.g., membrane separation.

Depending on the mass transfer capabilities of the system, the available quantity of hydrogen, the targeted product gas application, and/or to match the composition of the product gas with the available downstream purification options, it may be beneficial to tune the in-situ process, which can be facilitated using the contour maps in Figure 15. The produced process contours can be helpful to explore this. For example, a system with a mass transfer capability of 120 day$^{-1}$ may be optimised for 'near complete' in-situ biomethanation via operation at an OLR of 1 g VS L$^{-1}$ day$^{-1}$ (point B), with these conditions being able to produce a high-quality biomethane (~94% CH$_4$ ~5% CO$_2$, ~1% H$_2$) but only

at a low productivity (MER) of 0.1 day$^{-1}$ and while significantly underutilising the biomass treatment capacity of the system. Another option for the same system may be to accept a low biomethanation extent, but to operate at a reduced stoichiometric ratio (S = 0.25) and an increased OLR (Point C) to partially upgrade the biogas ($\sim$75% $CH_4$ $\sim$24% $CO_2$, $\sim$1% $H_2$) whilst minimising hydrogen contamination at a modest value of MER (0.75 L L$^{-1}$ day$^{-1}$). In another application, it may be better to maximise the consumption of carbon dioxide by applying a higher OLR and stoichiometric ratio (S = 0.9, Point D) to produce a hydrogen-rich biomethane blend ($\sim$65% $CH_4$ $\sim$10% $CO_2$, $\sim$25% $H_2$) which, upon $CO_2$ removal, could be suitable for natural gas grid injection depending on local requirements at an improved MER (0.85 L L$^{-1}$day$^{-1}$).

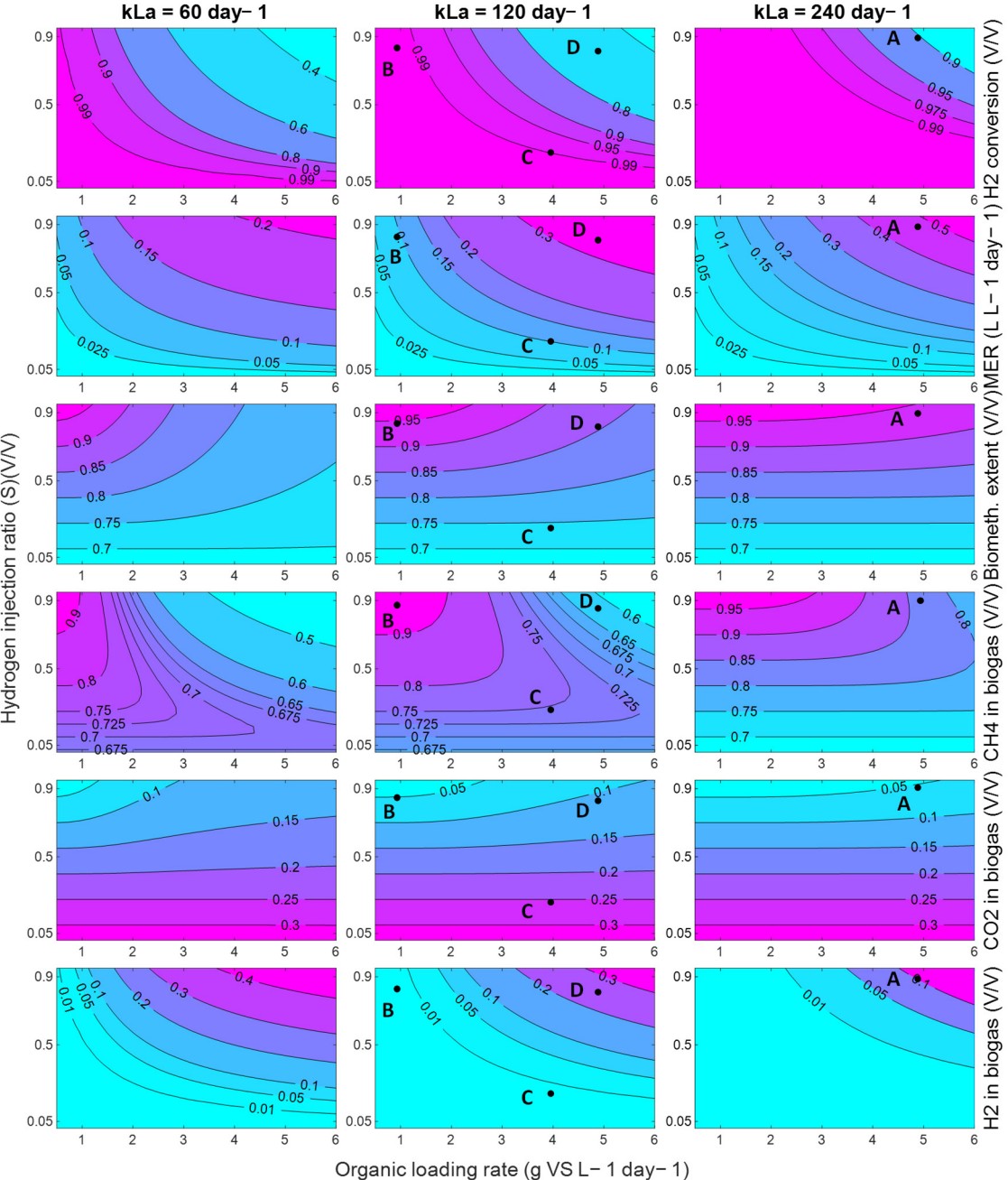

**Figure 15.** Contour plots of in-situ biomethanation performance using SS with a variation in the OLR and hydrogen injection ratio for $k_La$ of 60, 120, and 240 day$^{-1}$. Labelled points (A, B, C, D) on the plots refer to alternative process conditions and are elaborated in the discussion.

For reference, the equivalent contours for a FW fed system are shown in Appendix A.2 Figure A1. The trends are similar, but the challenge of producing both a high quality and quantity of biomethane product gas from an in-situ system is greater for FW due to its higher biogas production and $CO_2$ baseline content (and therefore greater hydrogen requirement).

## 4. Conclusions

In this work, in-situ biomethanation, alongside the AD of sewage sludge and food waste, was successfully demonstrated at laboratory scale. Continuous feedstock and hydrogen feeding as well as the monitoring and feedback control of the hydrogen supply were implemented to emulate the envisaged full-scale implementation of this technology. The complete results dataset is made available on a data repository [39].

The performance of the lab-scale process, in terms of its capability to produce a high-concentration biomethane output gas directly from the digesters, was generally limited, mainly due to the mass-transfer of gaseous hydrogen, which was evidenced by a large residual hydrogen concentration in the produced biogas, but otherwise the biological process was stable and showed no signs of process inhibition. Increasing the biogas recirculation rate, reducing the bubble size through a sparger, and increasing the mechanical mixing improved the gas–liquid mass transfer, with the $k_L a$ estimated to be between 43–82 day$^{-1}$. At an OLR of 2 g VS L$^{-1}$ day$^{-1}$, it was possible to achieve a MER of 0.17 and 0.23 L L$^{-1}$day$^{-1}$, a H$_2$ conversion of 80 and 66%, and a CH$_4$ content of 69 and 55% for SS and FW, respectively. The reduction in the OLR to 1 g VS L$^{-1}$ day$^{-1}$ allowed for an increase in biomethanation performance in terms of hydrogen conversion (94 and 87% for SS and FW, respectively) and methane content in biogas (79 and 68% for SS and FW, respectively), but at the expense of overall system productivity and the utilisation of the reactors.

To explore a broader set of operating conditions, a process model, based on a single AD reaction, the mass balance on the biomethanation reaction, and the simple treatment of gas–liquid mass transfer was developed and acceptably validated on experimental data. The exploration demonstrated the process design trade-offs that need to be made in order to have either a highly productive (high OLR and methane production) or a high-quality biomethane output gas (at low OLR) at the $k_L a$ values obtained in this experimental work. With the $k_L a$ increased up to >240 day$^{-1}$, a more complete in-situ biomethanation could be possible at OLR values common in large scale AD systems for SS, while for FW, even higher gas–liquid mass transfer rates would be required.

**Author Contributions:** Conceptualisation: A.S., D.P., S.M., and M.W; methodology; A.S., D.P., and M.W; modelling; D.P. and M.W; data-analysis: A.S., D.P., and M.W; writing—original draft preparation: A.S., D.P., and M.W; writing—review and editing: A.S., D.P., S.M., and M.W; supervision: W.N., D.P., M.P., and M.W.; project administration: W.N., D.P., M.P., and M.W; funding acquisition: W.N., M.P., and M.W. All authors have read and agreed to the published version of the manuscript.

**Funding:** This work was supported by the Indonesia Endowment Fund for Education (LPDP) (PhD Studentship) and the UK Engineering and Physical Sciences Research Council (EPSRC) through the IBCat H2AD project 'Biomethanisation of $CO_2$ in anaerobic digestion plants' (grant EP/M028208/1).

**Institutional Review Board Statement:** Not applicable.

**Informed Consent Statement:** Not applicable.

**Data Availability Statement:** Complete results dataset is available at the University of Sheffield data repository: https://doi.org/10.15131/shef.data.21747239.

**Conflicts of Interest:** The authors declare no conflict of interest.

## Nomenclature

Abbreviations

| | |
|---|---|
| AD | Anaerobic digestion |
| ADM1 | Anaerobic Digestion Model 1 |
| BMP | Biochemical methane potential |
| CSTR | Continuous stirred-tank reactor |
| FW | Food waste |
| FW1 | Duplicate 1 in food waste biomethanation experiments |
| FW2 | Duplicate 2 in food waste biomethanation experiments |
| HRT | Hydraulic retention time |
| IA | Intermediate alkalinity |
| ISR | Inoculum to substrate ratio |
| MER | Methane evolution rate (methane from biomethanation) [L L$^{-1}$ d$^{-1}$] |
| MFC | Mass flow controllers |
| O1,O2,O3 | Periods in the experimental stage at different operational conditions |
| OLR | Organic loading rate |
| PA | Partial alkalinity |
| R1,R2,R3 | Periods in the experimental stage at different recirculation rates |
| SS | Sewage sludge |
| SS1 | Duplicate 1 in sewage sludge biomethanation experiments |
| SS2 | Duplicate 2 in sewage sludge biomethanation experiments |
| STP | Standard temperature and pressure (273.15 K and 1 bar) |
| TAN | Total ammonia nitrogen |
| VS | Volatile solids |

Symbols

| | |
|---|---|
| $C_{H2,g}$ | Hydrogen concentration in the gas bulk phase [mol L$^{-1}$] |
| $C^{*}_{H2,l}$ | Dissolved hydrogen conc. in equilibrium with gas bulk phase [mol L$^{-1}$] |
| $C_{H2,l}$ | Dissolved hydrogen concentration in the liquid bulk phase [mol L$^{-1}$] |
| $G_{H2\_est}$ | Estimated stoichiometric hydrogen injection rate [mL min$^{-1}$] |
| $G_{H2\_CH4}$ | Scheduled hydrogen injection, based on CH$_4$ setpoint [mL min$^{-1}$] |
| $G_{H2\_CO2}$ | Scheduled hydrogen injection, based on CO$_2$ constraint [mL min$^{-1}$] |
| $G_{H2\_H2}$ | Scheduled hydrogen injection rate, based on H$_2$ constraint [mL min$^{-1}$] |
| $G_{H2\_pH}$ | Scheduled hydrogen injection rate, based on pH constraint [mL min$^{-1}$] |
| $G_{H2\_MFC}$ | Actual hydrogen injection flow rate, requested to the MFC [mL min$^{-1}$] |
| $H$ | Henry's dimensionless constant [(mol H2 L$^{-1}$)$_{gas}$/(mol H2 L$^{-1}$)$_{liq}$] |
| $\dot{n}_{G/L}$ | Molar gas–liquid transfer rate [mol d$^{-1}$] |
| $p$ | Gas pressure [bar] |
| $k_{CH4}$ | Gain parameter, of the scheduling control based on methane [-] |
| $k_{CO2}$ | Gain parameter, of the scheduling control based on carbon dioxide [-] |
| $k_{H2}$ | Gain parameter, of the scheduling control based on hydrogen [-] |
| $k_{pH}$ | Gain parameter, of the scheduling control based on pH [-] |
| $Q$ | Flow rate [L d$^{-1}$] |
| $Q_{biogas}$ | Total biogas outflow rate [L d$^{-1}$] |
| $Q_{CH_4,AD}$ | Methane flow rate from digestion of feedstock [L d$^{-1}$] |
| $Q_{CH_4,BM}$ | Methane flow rate from biomethanation [L$^{-1}$ d$^{-1}$] |
| $Q_{CO_2,AD}$ | Carbon dioxide flow rate from digestion of feedstock [L d$^{-1}$] |
| $Q_{H_2,dis}$ | Hydrogen dissolution rate [L$^{-1}$ d$^{-1}$] |
| $R$ | Ideal gas constant [L bar K$^{-1}$ mol$^{-1}$] |
| $RT_G$ | Gas retention time [hours] |
| $S$ | Hydrogen injection ratio (injected over stoichiometric requirement) [-] |
| $V_H$ | Overall gas phase volume of the system [L] |
| $V_R$ | Working volume, liquid phase, of biomethanation reactor [L] |
| $Y_{CH4}$ | Methane specific yield from digestion of feedstock [L g$^{-1}$ VS] |
| $Y_{CO2}$ | Carbon dioxide-specific yield from digestion of feedstock [L g$^{-1}$ VS] |
| $X_{H2}$ | Hydrogen conversion in biomethanation [-] |
| $\chi$ | Molar and volumetric gas fractions [-] |

## Appendix A.

*Appendix A.1. Results of Inoculum and Biomass Characterisation and Baseline AD Testing*

**Table A1.** Characterisation results of the inoculums used for BMP and in-situ biomethanation experiments.

| Inoculum | Unit | Sewage Sludge Inoculum | Food Waste Inoculum |
|---|---|---|---|
| TS | % | 3.04 | 3.97 |
| VS | % | 2.00 | 2.74 |
| pH | | 7.20 | 7.50 |
| Ammonia | gTAN/kg substrate | 1.43 | 6.42 |
| PA | $gCaCO_3$/kg | 3.26 | 18.40 |
| IA | $gCaCO_3$/kg | 1.24 | 6.13 |
| Total alkalinity | $gCaCO_3$/kg | 4.51 | 24.52 |
| IA/PA | | 0.38 | 0.33 |

**Table A2.** Characterisation results of the feedstocks used in the biomethanation experiments.

| Feedstock | Sewage Sludge | Food Waste |
|---|---|---|
| TS (% wet weight) | $3.96 \pm 0.55$ | $14.29 \pm 0.52$ |
| VS (% wet weight) | $2.79 \pm 0.43$ | $13.58 \pm 0.48$ |
| Elemental Analysis | | |
| Carbon (%TS) | $39.80 \pm 0.84$ | $49.92 \pm 0.31$ |
| Hydrogen (%TS) | $5.98 \pm 0.22$ | $6.77 \pm 0.77$ |
| Oxygen (%TS) | $26.58 \pm 1.51$ | $35.27 \pm 0.63$ |
| Nitrogen (%TS) | $4.80 \pm 0.56$ | $3.35 \pm 0.08$ |

**Table A3.** Results of baseline AD experiments across all reactors.

| Reactor | Methane (%) | Carbon Dioxide (%) | Specific Methane Yield (L g$^{-1}$ VS) | Specific Carbon Dioxide yield (L g$^{-1}$ VS) |
|---|---|---|---|---|
| Control SS | $65.38 \pm 0.01$ | $34.02 \pm 0.00$ | $0.25 \pm 0.13$ | $0.13 \pm 0.06$ |
| SS1 | $65.32 \pm 0.00$ | $34.26 \pm 0.00$ | $0.24 \pm 0.02$ | $0.12 \pm 0.01$ |
| SS2 | $64.97 \pm 0.01$ | $34.22 \pm 0.01$ | $0.23 \pm 0.06$ | $0.12 \pm 0.03$ |
| Control FW | $59.96 \pm 0.01$ | $39.62 \pm 0.01$ | $0.44 \pm 0.09$ | $0.29 \pm 0.05$ |
| FW1 | $59.34 \pm 0.01$ | $39.56 \pm 0.01$ | $0.42 \pm 0.05$ | $0.28 \pm 0.03$ |
| FW2 | $59.57 \pm 0.01$ | $39.90 \pm 0.01$ | $0.41 \pm 0.13$ | $0.27 \pm 0.07$ |

*Appendix A.2. In-Situ Biomethanation Results across All Experimental Periods*

**Table A4.** Average in-situ biomethanation results for sewage sludge (SS)-fed reactors across operational periods R1–R3.

| Experimental Period | | R1 | | | R2 | | | R3 | | |
|---|---|---|---|---|---|---|---|---|---|---|
| Reactor | | Control SS | SS1 | SS2 | Control SS | SS1 | SS2 | Control SS | SS1 | SS2 |
| Duration | days | 17 | 17 | 17 | 12 | 12 | 12 | 21 | 21 | 21 |
| OLR | gVS L$_R^{-1}$ day$^{-1}$ | 1.91 | 1.88 | 1.92 | 1.93 | 1.89 | 2.03 | 1.87 | 1.99 | 1.97 |
| H2 injection rate | L L$_R^{-1}$ day$^{-1}$ | | 0.81 | 0.80 | | 0.85 | 0.85 | | 0.85 | 0.87 |
| Specific biogas production | L gVS$^{-1}$ | 0.40 | 0.55 | 0.57 | 0.39 | 0.49 | 0.56 | 0.39 | 0.54 | 0.52 |
| Specific CH4 production | L gVS$^{-1}$ | 0.28 | 0.31 | 0.31 | 0.27 | 0.28 | 0.32 | 0.27 | 0.34 | 0.34 |
| Specific CO2 production | L gVS$^{-1}$ | 0.12 | 0.09 | 0.09 | 0.12 | 0.08 | 0.10 | 0.08 | 0.05 | 0.04 |
| H2 consumption rate | L L$_R^{-1}$ day$^{-1}$ | | 0.52 | 0.46 | | 0.61 | 0.56 | | 0.63 | 0.65 |
| H2 conversion | % | | 64.09 | 58.50 | | 71.41 | 66.32 | | 73.93 | 75.26 |
| Methane evolution rate (MER) | L L$_R^{-1}$ day$^{-1}$ | | 0.13 | 0.12 | | 0.15 | 0.14 | | 0.16 | 0.16 |

**Table A4.** *Cont.*

| Experimental Period | | R1 | | | R2 | | | R3 | | |
|---|---|---|---|---|---|---|---|---|---|---|
| Reactor | | Control SS | SS1 | SS2 | Control SS | SS1 | SS2 | Control SS | SS1 | SS2 |
| Gas retention time | hours | | 8.21 | 8.22 | | 9.42 | 7.63 | | 8.08 | 8.43 |
| Gas–liquid transfer rate $kLa$ | day$^{-1}$ | | 45.8 | 39.8 | | 55.8 | 52.5 | | 63.0 | 65.2 |
| CH4 content | vol.% | 69.91 | 56.16 | 54.29 | 68.50 | 59.47 | 58.80 | 68.64 | 63.65 | 64.72 |
| CO2 content | vol.% | 29.58 | 16.78 | 16.58 | 29.63 | 15.56 | 16.80 | 28.89 | 15.59 | 14.28 |
| H2 content | vol.% | 0.07 | 27.91 | 30.24 | 0.11 | 25.28 | 24.63 | 0.24 | 20.23 | 20.38 |
| pH | | 7.21 | 7.43 | 7.46 | 7.23 | 7.32 | 7.48 | 7.34 | 7.6 | 7.62 |
| TS | wt.% | 2.41 | 2.59 | 2.48 | 2.34 | 2.38 | 2.53 | 2.51 | 2.76 | 2.57 |
| VS | wt.% | 1.54 | 1.56 | 1.54 | 1.30 | 1.47 | 1.58 | 1.46 | 1.73 | 1.57 |
| Ammonia | gTAN L$^{-1}$ | 1.61 | 1.55 | 1.49 | 1.46 | 1.36 | 1.56 | 1.46 | 1.56 | 1.52 |
| Alkalinity ratio | | 0.45 | 0.42 | 0.40 | 0.37 | 0.38 | 0.31 | 0.33 | 0.31 | 0.31 |
| Acetate | g L$^{-1}$ | 0.25 | 0.25 | 0.08 | 0.38 | 0.25 | 0.12 | 0.68 | 0.23 | 0.21 |
| Total VFA | g L$^{-1}$ | 2.81 | 2.02 | 0.72 | 3.34 | 2.55 | 2.35 | 2.17 | 1.38 | 1.13 |

**Table A5.** Average in-situ biomethanation results for sewage sludge (SS)-fed reactors across operational periods O1–O3.

| Experimental Period | | O1 | | | O2 | | | O3 | | |
|---|---|---|---|---|---|---|---|---|---|---|
| Reactor | | Control SS | SS1 | SS2 | Control SS | SS1 | SS2 | Control SS | SS1 | SS2 |
| Duration | days | 19 | 19 | 19 | 20 | 20 | 10 | 18 | 18 | 18 |
| OLR | gVS L$_R^{-1}$ day$^{-1}$ | 1.94 | 1.85 | 1.79 | 2.12 | 2.07 | 2.10 | 0.88 | 0.88 | 0.99 |
| H2 injection rate | L L$_R^{-1}$ day$^{-1}$ | | 0.86 | 0.85 | | 0.85 | 0.72 | | 0.43 | 0.43 |
| Specific biogas production | L gVS$^{-1}$ | 0.43 | 0.50 | 0.47 | 0.29 | 0.37 | 0.28 | 0.27 | 0.4 | 0.36 |
| Specific CH4 production | L gVS$^{-1}$ | 0.30 | 0.32 | 0.33 | 0.21 | 0.24 | 0.19 | 0.20 | 0.31 | 0.29 |
| Specific CO2 production | L gVS$^{-1}$ | 0.14 | 0.07 | 0.06 | 0.08 | 0.05 | 0.04 | 0.07 | 0.05 | 0.05 |
| H2 consumption rate | L L$_R^{-1}$ day$^{-1}$ | | 0.66 | 0.70 | | 0.68 | 0.61 | | 0.40 | 0.41 |
| H2 conversion | % | | 76.68 | 82.82 | | 79.96 | 85.74 | | 92.53 | 95.55 |
| Methane evolution rate (MER) | L L$_R^{-1}$ day$^{-1}$ | | 0.16 | 0.18 | | 0.16 | 0.14 | | 0.10 | 0.10 |
| Gas retention time | hours | | 9.42 | 10.91 | | 11.89 | 19.75 | | 25.06 | 25.11 |
| Gas–liquid transfer rate $kLa$ | day$^{-1}$ | | 64.7 | 76.4 | | 66.1 | 69.3 | | 53.2 | 65.2 |
| CH4 content | vol.% | 67.90 | 65.36 | 71.49 | 69.92 | 66.22 | 72.05 | 71.1 | 77.62 | 80.90 |
| CO2 content | vol.% | 31.11 | 13.95 | 12.51 | 28.75 | 12.41 | 13.24 | 25.60 | 12.84 | 12.90 |
| H2 content | vol.% | 0.12 | 21.32 | 15.99 | 0.08 | 22.14 | 14.37 | 0.03 | 8.63 | 4.92 |
| pH | | 7.74 | 7.78 | 8.01 | 7.43 | 7.59 | 7.24 | 7.57 | 7.78 | 7.62 |
| TS | wt.% | 3.02 | 3.19 | 3.37 | 2.59 | 2.37 | 2.72 | 2.48 | 2.24 | 2.31 |
| VS | wt.% | 1.63 | 1.75 | 1.87 | 1.24 | 1.31 | 1.45 | 1.29 | 1.25 | 1.24 |
| Ammonia | gTAN L$^{-1}$ | 1.60 | 1.59 | 1.53 | 1.38 | 1.46 | 1.45 | 1.51 | 1.53 | 1.57 |
| Alkalinity ratio | | 0.41 | 0.42 | 0.38 | 0.17 | 0.41 | 0.63 | 0.14 | 0.14 | 0.15 |
| Acetate | g L$^{-1}$ | 0.94 | 2.11 | 2.94 | 0.95 | 2.00 | 2.25 | 0.31 | 0.23 | 0.40 |
| Total VFA | g L$^{-1}$ | 2.40 | 3.27 | 4.04 | 2.61 | 3.62 | 3.08 | 1.01 | 0.58 | 0.79 |

**Table A6.** Average in-situ biomethanation results for food waste (FW)-fed reactors across operational periods R1–R3.

| Experimental Period | | R1 | | | R2 | | | R3 | | |
|---|---|---|---|---|---|---|---|---|---|---|
| Reactor | | Control FW | FW1 | FW2 | Control FW | FW1 | FW2 | Control FW | FW1 | FW2 |
| Duration | days | 14 | 14 | 14 | 9 | 4 | 9 | 21 | 21 | 21 |
| OLR | gVS L$_R^{-1}$ day$^{-1}$ | 0.94 | 1.35 | 1.31 | 1.70 | 1.88 | 1.63 | 2.04 | 1.97 | 2.00 |
| H2 injection rate | L L$_R^{-1}$ day$^{-1}$ | | 0.8 | 0.795 | | 1.36 | 1.32 | | 1.46 | 1.49 |
| Specific biogas production | L gVS$^{-1}$ | 0.59 | 0.75 | 0.83 | 0.58 | 0.89 | 0.99 | 0.71 | 1.04 | 0.99 |

**Table A6.** *Cont.*

| Experimental Period | | R1 | | | R2 | | | R3 | | |
|---|---|---|---|---|---|---|---|---|---|---|
| Reactor | | Control FW | FW1 | FW2 | Control FW | FW1 | FW2 | Control FW | FW1 | FW2 |
| Specific CH4 production | L gVS$^{-1}$ | 0.38 | 0.40 | 0.42 | 0.36 | 0.42 | 0.48 | 0.45 | 0.50 | 0.53 |
| Specific CO2 production | L gVS$^{-1}$ | 0.20 | 0.17 | 0.19 | 0.21 | 0.18 | 0.17 | 0.24 | 0.18 | 0.18 |
| H2 consumption rate | L L$_R^{-1}$ day$^{-1}$ | | 0.56 | 0.51 | | 0.82 | 0.79 | | 0.81 | 0.94 |
| H2 conversion | % | | 70.64 | 64.66 | | 60.02 | 59.71 | | 56.26 | 63.7 |
| Methane evolution rate (MER) | L L$_R^{-1}$ day$^{-1}$ | | 0.13 | 0.12 | | 0.20 | 0.20 | | 0.19 | 0.22 |
| Gas retention time | hours | | 11.12 | 10.15 | | 7.83 | 5.29 | | 5.43 | 4.57 |
| Gas–liquid transfer rate $kLa$ | day$^{-1}$ | | 53.8 | 46.9 | | 67.8 | 65.3 | | 67.9 | 83.0 |
| CH4 content | vol.% | 63.83 | 52.75 | 50.86 | 62.45 | 47.96 | 49.67 | 61.82 | 48.45 | 53.18 |
| CO2 content | vol.% | 35.09 | 24.23 | 23.82 | 36.33 | 20.26 | 17.67 | 36.36 | 19.56 | 18.64 |
| H2 content | vol.% | 0.10 | 23.13 | 25.50 | 0.12 | 32.82 | 33.56 | 0.16 | 32.11 | 28.15 |
| pH | | 7.6 | 7.66 | 7.63 | 7.71 | 7.86 | n.a | 7.77 | 7.94 | 8.16 |
| TS | wt.% | 3.71 | 3.78 | 3.63 | 3.24 | 3.24 | 2.97 | 4.08 | 3.50 | 4.34 |
| VS | wt.% | 2.73 | 2.80 | 2.73 | 2.38 | 2.29 | 2.24 | 2.82 | 2.46 | 3.14 |
| Ammonia | gTAN L$^{-1}$ | 4.42 | 4.25 | 4.13 | 3.80 | 3.62 | 3.50 | 3.66 | 3.69 | 3.60 |
| Alkalinity ratio | | 0.63 | 0.77 | 0.78 | 0.39 | 0.35 | 0.34 | 0.33 | 0.29 | 0.28 |
| Acetate | g L$^{-1}$ | 0.64 | 1.31 | 1.53 | 0.47 | 0.53 | 0.51 | 0.37 | 0.34 | 0.48 |
| Total VFA | g L$^{-1}$ | 1.30 | 2.07 | 2.26 | 0.80 | 0.86 | 0.83 | 0.89 | 0.86 | 0.98 |

**Table A7.** Average in-situ biomethanation results for food waste (FW)-fed reactors across operational periods O1–O3.

| Experimental Period | | O1 | | | O2 | | | O3 | | |
|---|---|---|---|---|---|---|---|---|---|---|
| Reactor | | Control FW | FW1 | FW2 | Control FW | FW1 | FW2 | Control FW | FW1 | FW2 |
| Duration | days | 32 | 32 | 32 | 20 | 20 | 20 | 18 | 18 | 18 |
| OLR | gVS L$_R^{-1}$ day$^{-1}$ | 1.79 | 2.01 | 2.06 | 2.05 | 1.64 | 1.71 | 0.95 | 0.97 | 0.99 |
| H2 injection rate | L L$_R^{-1}$ day$^{-1}$ | | 1.47 | 1.51 | | 1.38 | 1.43 | | 0.74 | 0.77 |
| Specific biogas production | L gVS$^{-1}$ | 0.69 | 1.12 | 1.06 | 0.61 | 1.00 | 1.00 | 0.78 | 0.78 | 0.75 |
| Specific CH4 production | L gVS$^{-1}$ | 0.43 | 0.52 | 0.53 | 0.38 | 0.52 | 0.55 | 0.54 | 0.51 | 0.53 |
| Specific CO2 production | L gVS$^{-1}$ | 0.26 | 0.23 | 0.22 | 0.24 | 0.18 | 0.18 | 0.24 | 0.15 | 0.14 |
| H2 consumption rate | L L$_R^{-1}$ day$^{-1}$ | | 0.73 | 0.84 | | 0.90 | 0.96 | | 0.64 | 0.69 |
| H2 conversion | % | | 49.65 | 56.06 | | 64.99 | 67.62 | | 85.11 | 89.82 |
| Methane evolution rate (MER) | L L$_R^{-1}$ day$^{-1}$ | | 0.18 | 0.21 | | 0.22 | 0.24 | | 0.16 | 0.17 |
| Gas retention time | hours | | 3.94 | 3.99 | | 6.67 | 5.08 | | 12.06 | 12.33 |
| Gas–liquid transfer rate $kLa$ | day$^{-1}$ | | 58.9 | 71.9 | | 78.0 | 86.8 | | 72.5 | 88.1 |
| CH4 content | vol.% | 61.96 | 46.79 | 50.20 | 61.54 | 54.50 | 56.05 | 67.37 | 66.26 | 70.56 |
| CO2 content | vol.% | 37.48 | 20.94 | 20.45 | 37.97 | 16.95 | 18.03 | 31.84 | 19.51 | 18.93 |
| H2 content | vol.% | 0.12 | 34.96 | 30.45 | 0.05 | 29.45 | 26.80 | 0.02 | 14.04 | 10.23 |
| pH | | 7.75 | 8.18 | 8.10 | 7.76 | 8.20 | 8.20 | 7.69 | 8.13 | 8.25 |
| TS | wt.% | 3.04 | 2.83 | 2.92 | 2.91 | 2.78 | 2.74 | 2.72 | 2.54 | 2.55 |
| VS | wt.% | 2.11 | 1.99 | 2.07 | 2.00 | 1.92 | 1.90 | 1.84 | 1.65 | 1.75 |
| Ammonia | gTAN L$^{-1}$ | 3.84 | 3.95 | 3.87 | 4.10 | 3.86 | 4.14 | 4.32 | 3.45 | 4.14 |
| Alkalinity ratio | | 0.31 | 0.30 | 0.28 | 0.29 | 0.28 | 0.21 | 0.61 | 0.14 | 0.21 |
| Acetate | g L$^{-1}$ | 0.84 | 0.84 | 0.88 | 2.01 | 1.86 | 0.65 | 4.42 | 0.61 | 0.35 |
| Total VFA | g L$^{-1}$ | 1.19 | 1.15 | 1.17 | 3.11 | 3.02 | 1.12 | 6.88 | 1.20 | 0.74 |

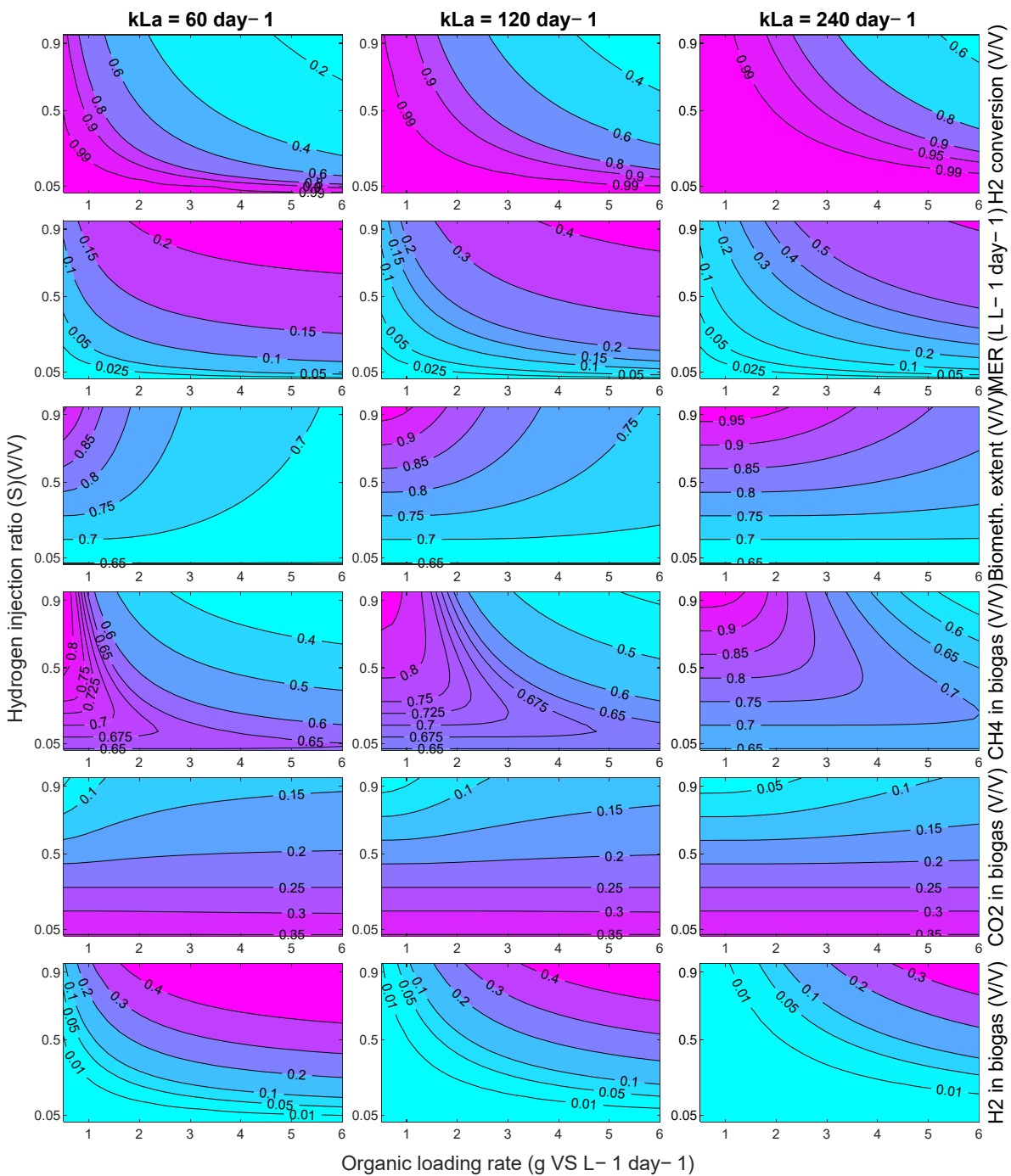

**Figure A1.** Contour plots of modelled in-situ biomethanation performance using FW with a variation in the OLR and hydrogen injection ratio for $k_La$ of 60, 120, and 240 $day^{-1}$.

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
