# Peer review of "Experimental Evaluation of Continuous In-Situ Biomethanation of CO2 in Anaerobic Digesters Fed on Sewage Sludge and Food Waste and the Influence of Hydrogen Gas–Liquid Mass Transfer"

_processes, doi:10.3390/pr11020604_

Round 1

Reviewer 1 Report

An automated rig with a monitoring and control system was applied to the research work on continuous in-situ biomethanation in anaerobic digesters. The paper is well written with clear structure and logic flow. Apart from direct experimental observation, modelling work is also conducted to further investigate the process and the required mass transfer rate for H2 is therefore identified. 

A few minor issues, however, should be taken into consideration during revision:

- Lines 2-3: Is 'biomethanation of hydrogen' an accurate description of the process? The conventional term is either biomethanation of CO2 or biomethanation of CO2 and H2.

- A list of Nomenclature shall be provided considering the number of symbols used in the paper. 

- Lines 21-23: The operating conditions, e.g. OLR, should be provided when describing the effect of recirculation and stirring rate on methane evolution rate and He conversion, otherwise the results cannot be evaluated properly.

- Line 142 and throughout the paper: Please correct the cross reference error.

- Figure 1: Please explain 'SCADA'.

- Figures 1 and 2 can be coordinated better. For instance, it would be good to ensure all the units shown in Figure 1 are labelled and highlighted in Figure 2 to distinguish them from the additional labels in Figure 2. 

- Line 189: Please identify whether 'each experimental run' represents each experimental period (line 222) or each stage (line 224). 

- Table 3: Please add hydraulic retention time to the table.

- Line 242: Please ensure the symbols used are consistent throughout the paper. For instance, the symbol for reactor working volume VH is changed to VR (Line 257) or Vr (equation 6), and total biogas outflow (Qbiogas) is defined as total biogas flow later (Line 314). 

- Equation 11: Please check this equation carefully. It is not clear why a molar transfer rate nG/L can be replaced by a calculation for volume. 

- Equation 19: QCH4 should be one quarter of QH2, rather than 4QH2.

- Figure 7 and following relevant figures: Flow should be presented in units of NL/L-day.  

- Figure 9: It is not clear in what units the gas retention time is presented. 

- Table A2: the TS of food waste does not match the value given in the main text (line 104).

- Tables B1-B4: Average hydrogen injection and hydrogen consumption rate should be presented in units of L/L-day. Please also identify what material the average retention time refers to, e.g. gas or organic feed. 

Reviewer 2 Report

This paper is recommended for publication in the Processes journal after a major revision according to the following comments:

1- The manuscript should be reviewed and the phrase "Error! Reference source not found." should be removed throughout the text.

2- Conclusion is too long. It is better to be shortened (maximum three paragraphs).

3- All references must be checked. For example: Reference number 4; the abbreviation of the journal title "Journal of Fermentation Technology" is "J Ferment Technol". Also, in the References section: "H2", "CO2", and "CH4" should be "H2", "CO2", and "CH4". 

Reviewer 3 Report

This is a very well written manuscript, in fact one of the best that I have received to review in the last years. The Introduction is short but complete, understandable and points out the necessity of the work and the open questions to be answered. The aims of the investigations and experiments are clearly defined.

Material and methods are fully explained as well as the mathematical appoach for data evaluation and processing. Results are presented in a clear and understandable way.

Furthermore, the topic of the manuscript is of hig scientific relevance and important scientific questions could be answered by this work.

The only remark from my side is that the cross references in the manuscript were not updated, but I think this will be checked anyway by the publishing department.

Round 2

Reviewer 2 Report

The authors have revised the manuscript follow the comments, and I suggest to accept the manuscript in its current form. 

Author Response

Thank you